# Cryo-EM structure of the botulinum neurotoxin A/SV2B complex and its implications for translocation

Basavraj Khanppnavar [1,3], Oneda Leka[1,3], Sushant K. Pal [1],
Volodymyr M. Korkhov [1,2] ✉ & Richard A. Kammerer [1] ✉

Botulinum neurotoxin A1 (BoNT/A1) belongs to the most potent toxins and is used as a major therapeutic agent. Neurotoxin conformation is crucial for its translocation to the neuronal cytosol, a key process for intoxication that is only poorly understood. To gain molecular insights into the steps preceding toxin translocation, we determine cryo-EM structures of BoNT/A1 alone and in complex with its receptor synaptic vesicle glycoprotein 2B (SV2B). In solution, BoNT/A1 adopts a unique, semi-closed conformation. The toxin changes its structure into an open state upon receptor binding with the translocation domain ($H_N$) and the catalytic domain (LC) remote from the membrane, suggesting translocation incompatibility. Under acidic pH conditions, where translocation is initiated, receptor-bound BoNT/A1 switches back into a semi-closed conformation. This conformation brings the LC and $H_N$ close to the membrane, suggesting that a translocation-competent state of the toxin is required for successful LC transport into the neuronal cytosol.

Botulinum neurotoxins (BoNTs) are the most potent biological toxins[1–4]. They are divided into seven classical serotypes, designated BoNT/A–BoNT/G, and the recently identified BoNT/X that are all mainly produced by the gram-positive bacterium *Clostridium botulinum*[1–5]. The number of BoNTs is further increased by more than 40 different subtypes within serotypes A, B, E, and F[6]. Although many of these BoNT subtypes remain to be characterized in detail, some differ from the other toxins within the same serotype with respect to their catalytic properties, substrate specificity, duration of action, and efficiency to enter neuronal cells[7–11]. Furthermore, three non-clostridial BoNT-like proteins, termed BoNT/Wo, BoNT/En, and PMP1, have recently been identified in *Weissella oryzae SG25T*[12], *Enterococcus faecium*[13] and *Paraclostridium bifermentans*[14], respectively. Notably, PMP1 is specific for insects while the targets of BoNT/Wo and BoNT/En are unknown.

BoNTs can cause botulism in vertebrates, a rare paralytic and potentially life-threatening disease[15,16]. As a result, these toxins represent some of the most dangerous biological weapons known[17,18]. However, the extreme toxicity of BoNTs is matched by their usefulness in a wide range of clinical applications, which places these toxins among the most widely used therapeutic proteins. Currently, cosmetic and clinical applications are predominantly limited to the use of BoNT/A subtype 1 (BoNT/A1)[2,19,20].

BoNTs and BoNT-like proteins share a common domain organization and are produced as precursors that are cleaved into a 50 kDa light chain (LC) and a 100 kDa heavy chain (HC) that with the exception of BoNT/Wo remain connected by a disulfide bond[1,5,21–23]. The LC is a zinc metalloprotease that specifically cleaves soluble N-ethylmaleimide-sensitive-factor attachment receptor (SNARE) proteins. SNARE cleavage by BoNTs inactivates the vesicular fusion machinery within presynaptic nerve terminals, which blocks the release of the neurotransmitter acetylcholine at the neuromuscular junction and thus leads to a flaccid paralysis of muscles[16]. This paralysis underlies the adverse effects of botulism as well as the therapeutic applications of BoNTs. Serotypes BoNT/A and E cleave synaptosomal-associated protein 25 (SNAP-25), BoNTs B, D, F, G, Wo, and X cut vesicle-associated membrane protein (VAMP), and PMP1 cleaves syntaxin. BoNT/C and BoNT/En can cleave two

---

[1]PSI Center for Life Sciences, Villigen, Switzerland. [2]Institute of Molecular Biology and Biophysics, ETH Zurich, Zurich, Switzerland. [3]These authors contributed equally: Basavraj Khanppnavar, Oneda Leka. ✉e-mail: volodymyr.korkhov@psi.ch; richard.kammerer@psi.ch

substrates, SNAP-25 and syntaxin, and SNAP-25 and VAMP, respectively[1,5,12–14].

The HC comprises a 50-kDa N-terminal translocation domain ($H_N$) and a similarly-sized C-terminal receptor-binding domain ($H_C$). A dual-interaction mechanism involving two different receptors has been shown to be required for specific uptake of BoNTs into neuronal cells[24,25]. Most BoNTs bind to a polysialoganglioside (PSG) and a protein receptor. BoNTs B and G bind to synaptotagmin (Syt) and BoNTs A, D, E, and F bind synaptic vesicle glycoprotein 2 (SV2)[1,4,26]. Recent evidence suggests that BoNT/A1 even requires a tripartite PSG-Syt1-SV2 complex for entry into synaptic vesicles[27]. The cell-surface receptors of BoNT/X, BoNT/En, PMP1, and BoNT/Wo have not yet been identified, although the first three proteins contain a predicted PSG-binding site. Furthermore, a lipid-binding loop was recently identified in BoNTs B, C, D, DC, and G that appears essential for the potency of the toxins[28]. A lipid-binding site was also identified in the N-terminal half of $H_CA$[29]. Moreover, it was shown that N-glycosylation of SV2 contributes to BoNT/A binding[30]. A multiple interaction model with receptors or particular posttranslational modifications that have moderate toxin affinity would provide a plausible explanation why BoNTs are so extremely toxic at very low concentrations and why they are highly specific for predominantly cholinergic nerve terminals.

Upon binding to their respective receptors on the neuronal cell surface, BoNTs are imported into synaptic vesicles by receptor-mediated endocytosis (Fig. 1)[1,31]. The lipid environment and the acidification of synaptic vesicles are believed to trigger poorly understood conformational changes in $H_N$ that lead to the formation of an ion-conductive transmembrane channel through which the LC is shuttled across the membrane into the cytosol. In this process, a conserved hydrophobic viral-fusion-peptide-like sequence in $H_N$ (E620-F667) that gets surface exposed at low pH might mediate the initial step of $H_N$ insertion into the synaptic vesicle membrane[32]. Once reaching the cytosolic surface on the synaptic vesicle membrane, the disulfide bond connecting LC and HC is reduced by the thioredoxin-thioredoxin reductase system, resulting in the release of the protease[33] (Fig. 1).

The three SV2 paralogs, SV2A, SV2B, and SV2C[34–36], and the two synaptic vesicle 2-related proteins, SVOP and SVOPL, belong to a subgroup of the solute carrier 22 (SLC22) family[37]. Also known as the organic ion transporter family, it consists of 28 members that phylogenetically cluster together based on their specificity for organic cations, organic anions, and organic zwitterions/cations[37]. SV2 proteins differ from the other family members by their unique, large luminal domain (LD) between transmembrane segments 7 and 8 (missing in SVOP and SVOPL)[38–40]. Although the precise physiological functions of SV2 proteins remains to be elucidated, functional studies

suggest several roles of SV2 proteins in vesicular function, including: vesicular transport, assistance of vesicular neurotransmitter loading, anchorage for vesicle structure and cycling, regulation of vesicular calcium sensitivity and interactions with the extracellular matrix[38–40]. The SV2 family is highly relevant to human health and disease. SV2A was identified as a specific target for the antiepileptic drug levetiracetam[41]. Furthermore, SV2 proteins have been linked to schizophrenia[42] and neurodegenerative disorders such as Alzheimer's disease, Parkinson's disease, and Huntington's disease[38–40], although the precise role of SV2 in these diseases is not clear.

While the physiological functions of SV2 proteins remains to be determined, their best-established role is to act as receptors for several BoNTs (BoNT/A, /D, /E, and /F)[43–46]. The binding site of BoNT/A has been mapped to the luminal domain (LD) region of SV2[45]. For SV2C it has been shown that protein-protein interactions are sufficient for stable complex formation with BoNT/A[30,47–49]. In contrast, N-linked glycosylation appears to play an important role in the interaction of SV2A and SV2B with the toxin[45]. However, structural information on toxin-receptor complexes is restricted to isolated domains[26,30,47–50]. A structure of a complex between a full-length receptor bound to a full-length BoNT has not been determined until now. This limits our understanding of the molecular principles that underlie BoNT recognition and translocation. A deep understanding of the molecular mechanisms of neurotoxin recognition and translocation is a prerequisite for the development of more effective BoNT-based therapies.

In this work, we provide fundamental molecular insights into the steps preceding toxin translocation. We determine cryo-EM structures of BoNT/A1 alone and in complex with its receptor SV2B. We show that in solution BoNT/A1 has a unique, semi-closed conformation, which switches its structure into an open state upon receptor binding with the $H_N$ and LC pointing away from the membrane. This observation suggests that this conformation of the toxin prevents translocation. Under acidic pH conditions, receptor-bound BoNT/A1 reverts back to a semi-closed conformation, which brings its LC and $H_N$ close to the membrane. This finding suggests that the toxin adopts a translocation-competent state at low pH.

## Results

### In solution, BoNT/A1 adopts a conformation distinct from that in the crystal

Cryo-EM analysis of full-length toxin proteins holds the promise to significantly contribute to the understanding of the complex mechanism of BoNT action as it enables direct insights into the dynamic interplay of toxin conformations that seem to play a fundamental role in BoNT translocation[31]. As yet, however, the cryo-EM

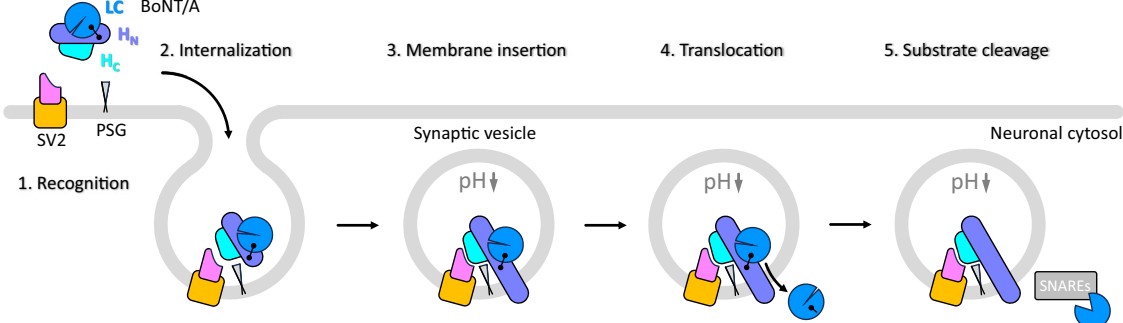

**Fig. 1 | Schematic representation of BoNT/A1 receptor recognition and translocation across the synaptic vesicle membrane.** Five distinct stages of neurotoxin mediated intoxication are illustrated: receptor recognition, internalization by receptor mediated-endocytosis, membrane insertion of $H_N$ and transmembrane channel formation, LC translocation and reduction of the disulfide bride connecting the LC and HC on the cytosolic face of the synaptic vesicle membrane by the thioredoxin-thioredoxin reductase system, substrate cleavage. The domains of BoNT/A1 are colored as follows: LC, blue; $H_N$, light blue; $H_C$, cyan; the LD domain and the TM region of SV2B are shown in pink and orange, respectively. The disulfide bridge connecting the LC and HC of BoNT/A1 is indicated. The polysialoganglioside (PSG) is shown as an anchored triangular shape.

structures of only three BoNTs, BoNT/B, BoNT/E, and BoNT/X, have been reported[51,52], emphasizing the need for new structural studies. For BoNT/A1, X-ray crystal structures of the full-length toxin have been determined[21]. However, as crystal packing interactions may induce non-natural arrangements of individual domains, elucidation of BoNT/A1 structures in solution is highly desirable.

To obtain a more complete picture of toxin conformations, we set out to solve the structure of full-length BoNT/A1 by single-particle cryo-EM. Towards this goal, an inactive variant was recombinantly produced in *E. coli* and purified to homogeneity (Supplementary Fig. 1A, B). We prepared cryo-EM grids containing the toxin in vitreous ice and subjected the sample to extensive cryo-EM data collection and image analysis (Supplementary Fig. 2). Particle sizes obtained for BoNT/A1 were consistent with the presence of a monomer (Supplementary Fig. 1B). The best subset of particles yielded a cryo-EM density at an overall resolution of 3.8 Å (Fig. 2a, b, Supplementary Figs. 2, 3A, 4A and Supplementary Table 1).

Based on our 2D and 3D classification, the semi-closed toxin state is the dominant species, although BoNT/A1 appears to sample a wide range of $H_C$ domain orientations in solution, including the open conformation (Supplementary Fig. 2). The $H_C$ density in our best 3D class for the BoNT/A1 in solution is relatively poorly defined, which reflects the previously reported high mobility of the $H_C$ (Fig. 2a, b and Supplementary Fig. 5)[51]. The poor density allowed only manual rigid-body docking of the domain into the semi-closed orientation. Key elements of the $H_N$ domain that have been previously suggested to play an important role in toxin translocation, such as the switch region (E620-F667)[32] and most of the belt (N493-D546) are relatively well resolved despite the moderate overall resolution of the cryo-EM map (we omitted the first two residues of the belt due to weak density; Supplementary Fig. 5).

The conformation of BoNT/A1 seen in the cryo-EM structure markedly differs from the open conformation observed in the X-ray crystal structure (Fig. 2c)[21]. Superimposition of the cryo-EM and the X-ray crystal structure of BoNT/A1 demonstrates that the LC and the $H_N$ domain arrangement is very similar and that the two structures basically differ by a ~107° rotation of the $H_C$ domain. The state of BoNT/A1 in solution captured by cryo-EM is reminiscent of the semi-open conformation of the closely-related tetanus toxin at pH 5.5–6.5 (Fig. 2d)[53]. It is also similar to the conformation of BoNT/A1 predicted by AlphaFold, apart from an orientation of $H_C$ that differs by a rotation of ~180° (Fig. 2e). This domain arrangement is superficially also reminiscent of the BoNT/A1 conformation in the minimal progenitor toxin complex (M-PTC) with non-toxic nonhemagglutinin (NTNHA)[54] (see also Fig. 7d and 8a). However, the arrangement of BoNT/A1 domains markedly differs from the one seen in the closed conformations of BoNT/E1[22] and TeNT[53] (Fig. 2d, f).

## Cryo-EM structure of the SV2B-BoNT/A1 complex

Despite the central role of SV2 proteins in BoNT intoxication, high-resolution structural information on the full-length receptor complexes with full-length neurotoxins is missing. To bridge this gap in our knowledge, we transiently produced all three SV2 isoforms in FreeStyle 293 (HEK293F) cells. SDS-PAGE analysis of the SV2 proteins revealed single bands that migrated near the expected molecular weight (Supplementary Fig. 1C). The protein bands appeared somewhat blurred, which is likely due to the N-glycosylation of SV2 proteins in the LD region. SDS-PAGE analysis and SEC profiles indicated the best sample quality for SV2B (Supplementary Fig. 1C–E), followed by SV2A and SV2C. This conclusion was confirmed by negative-staining TEM (Supplementary Fig. 1J–L), which revealed a homogeneous distribution of similarly sized particles of SV2B, making this protein the focus of our further experiments.

SV2 proteins interact with several BoNTs through their LD[43–46]. For SV2C, it has been shown that protein-protein interactions are sufficient

for stable complex formation with BoNT/A[30,47–49]. For SV2A and SV2B, receptor the contribution of N-glycosylation appears to be more important for the interaction with the toxin[45]. To elucidate how SV2B interacts with the toxin, we prepared SV2B complexes with the inactive full-length BoNT/A1 (in the presence of a 20 molar excess of ganglioside GT1b) and $H_C$A1. Both samples were subjected to extensive cryo-EM data collection and image processing (Supplementary Figs. 3B–D, 4B, C, 6–8 and Supplementary Table 1). Although applied in excess, GT1b was not detectable in the complex structure with the full-length toxin. The resulting refined 3D reconstructions of SV2B-$H_C$A1 and SV2B-BoNT/A1 complexes were obtained at 3.7 Å and 4 Å resolution, respectively (Fig. 3a–c, Supplementary Fig. 3B–D, 4B, C and Supplementary Table 1). While we were submitting this manuscript, the cryo-EM structures of SV2A and SV2B were reported[55,56]. Overall, they are very similar to our structure. A comparison of our SV2B-$H_C$A1 cryo-EM structure to the published receptor structures is shown in the Supplementary Fig. 9.

The SV2B-BoNT/A1 and SV2B-$H_C$ structures revealed that the $H_C$ domain binds to the LD of SV2B, extending one β-sheet of the β-helix fold by the antiparallel β-hairpin located at the convex side between the two toxin $H_C$ subdomains (Fig. 3b). The overall architecture of the two complexes is very similar (RMSD 0.26 Å over 544 aligned residue pairs). The structures are also similar to the previously published $H_C$A-SV2C-LD complex structure[30,47,48] with RMSD values of 1.3 and 1.5 Å over 408 aligned residue pairs in SV2B-$H_C$A1 and SV2B-BoNT/A1, respectively, when compared to the representative structure PDB4JRA.

## Molecular architecture of SV2B

Our attempts to obtain a structure of SV2B alone proved unsuccessful. Our description of the SV2B structure is therefore based on the cryo-EM reconstruction of the SV2B-$H_C$A1 complex, which features a very well resolved density for the complete TM region of SV2B (Supplementary Figs. 6 and 8). A substantial portion of the receptor spanning amino-acid residues D87 to Q345 and R362 to E679 could be resolved by cryo-EM, confirming that the N-terminus of the protein is unstructured (Fig. 3d). The overall SV2 fold consists of three parts: the LD, the transmembrane domain (TM) and the intracellular structure comprising four helices (Fig. 3d). A Dali search[57] indicated that the proteins that are most structurally similar to SV2B in the Protein Data Bank (PDB) are organic cation transporters (OCTs) that also belong to the SLC22 family[37]. Notably, OCTs also share a similar arrangement of domains: an extracellular domain, a TM domain, and an intracellular helical bundle-like structure[58–60]. Our SV2B structures revealed a classical MFS fold for the SV2B TM part, with the putative transporter module composed of twelve TM helices (TM1-12). By analogy with other MFS transporter family members, the putative translocation pathway is located at the interface of the two-fold pseudo-symmetrically related transmembrane segments consisting of TM1-6 and TM7-8 (Figs. 3b and 4a, b). Interestingly, like OCTs, the putative substrate translocation pathway of SV2B encompasses negatively charged-residues (D122 and D611), suggesting it might also transport positively charged substrates. In our experimental structure, we also observe undefined density in proximity of D122 and D611, which could be positively-charged ions (Supplementary Fig. 10E). This region overlaps with canonical substrate-binding site in OCTs (Supplementary Fig. 10F).

The most prominent feature of SV2B is the unique and characteristic LD located between TM7 and TM8 (Fig. 3). It is well resolved and stably connected to TM7 and TM8. The conserved cysteine in the LD region, C526, forms a disulfide bridge with C141 in the luminal loop 1 (LL1) (Figs. 3d, 4a and Supplementary Fig. 10A–D). These Cys residues are conserved among SV2 proteins (Supplementary Fig. 10D) and may represent an important and functionally-relevant motif in all members of this family.

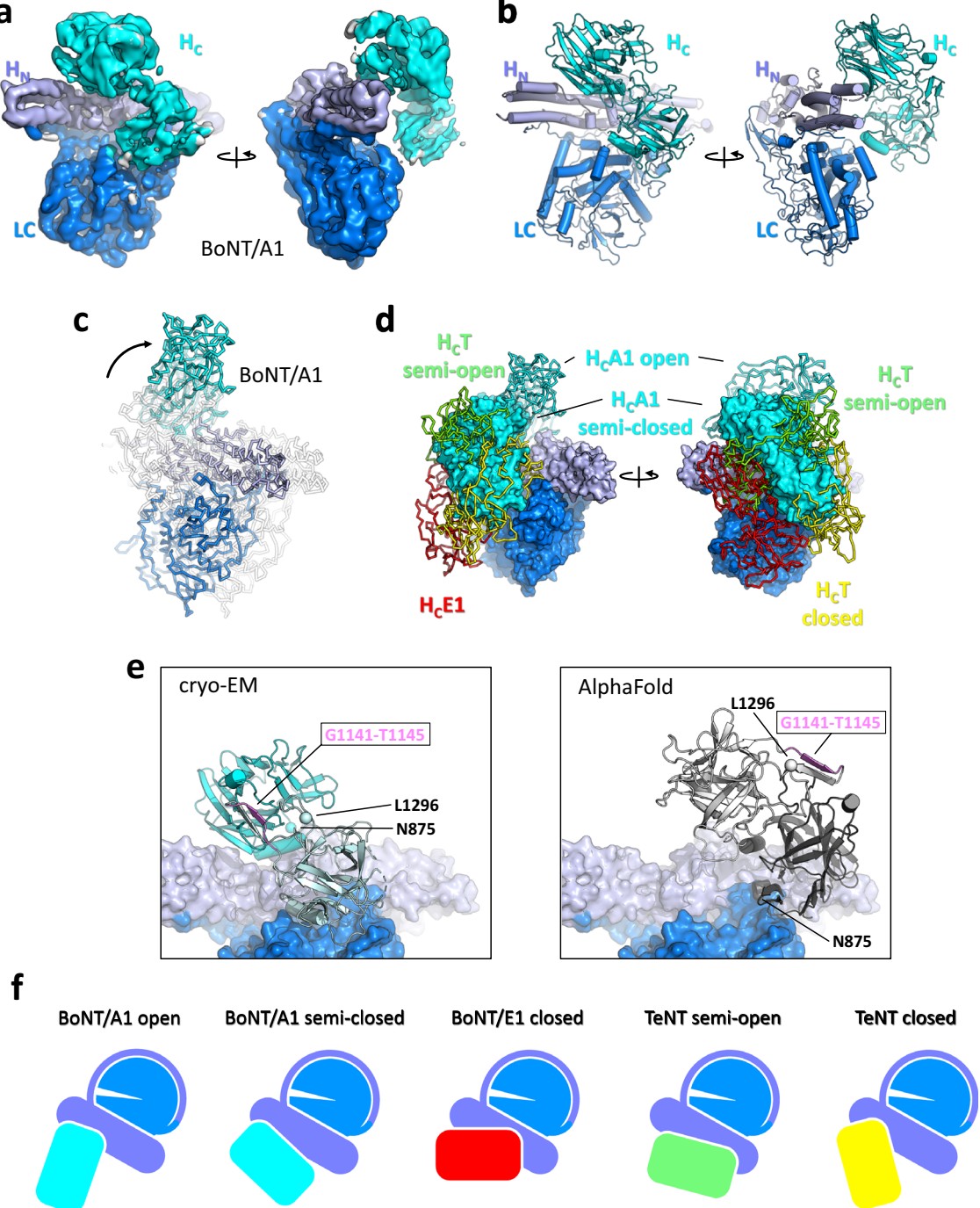

**Fig. 2 | Cryo-EM structure of BoNT/A1. a, b** Structure of BoNT/A1 at 3.8 Å resolution reveals a semi-closed conformation at physiological pH. **c** Comparison of the cryo-EM and X-ray crystal (PDB3BTA) structures of BoNT/A1, illustrating the conformational change of the H$_C$ in the toxin. **d** Superposition of the cryo-EM structure of BoNT/A1 with the X-ray crystal structure of BoNT/A1 (PDB3BTA), the cryo-EM structure of BoNT/E (PDB7QFP), X-ray crystal structure of TeNT (PDB5N0B) and cryo-EM structure of TeNT (EMD-3588). **e** Comparison of the BoNT/A1 cryo-EM structure with the structure predicted by AlphaFold (AF-P0DPI1-F1). One β-strand of H$_C$A1 at the toxin-receptor interface (residues G1141-T1145) is colored pink, illustrating the differences in H$_C$ domain orientation. **f** Illustration of the conformational plasticity seen in clostridium toxins.

Like SV2C-LD, SV2B-LD adopts a quadrilateral β-helix fold characteristic of pentapeptide-repeat proteins with every fifth amino acid being a hydrophobic residue[47]. Each of the four sides of SV2C-LDs consist of parallel β-sheets (Figs. 3, 5a and 6a−c). The structure is stabilized through a hydrophobic core that consists mainly of stacked hydrophobic phenylalanine side chains.

Comparison of the cryo-EM structure of SV2B with the AlphaFold-generated model revealed a high degree of overall similarity (RMSD of 1.54 between the two models) (Fig. 3e). However, substantial differences are seen in the C-terminal extension of LD, the position of LL1 and in the conformation of TM1-2 (Fig. 4a). Displacement of these loops in the predicted model results in the opening of the putative transporter translocation pathway, leading to an "outward-open" state (Fig. 4a, b), analogous to the state seen in other MFS transporters, including the SLC22 family members such as OCT1[59,60] or OCT3[58] (Fig. 4c, d). In contrast, the experimentally determined cryo-EM

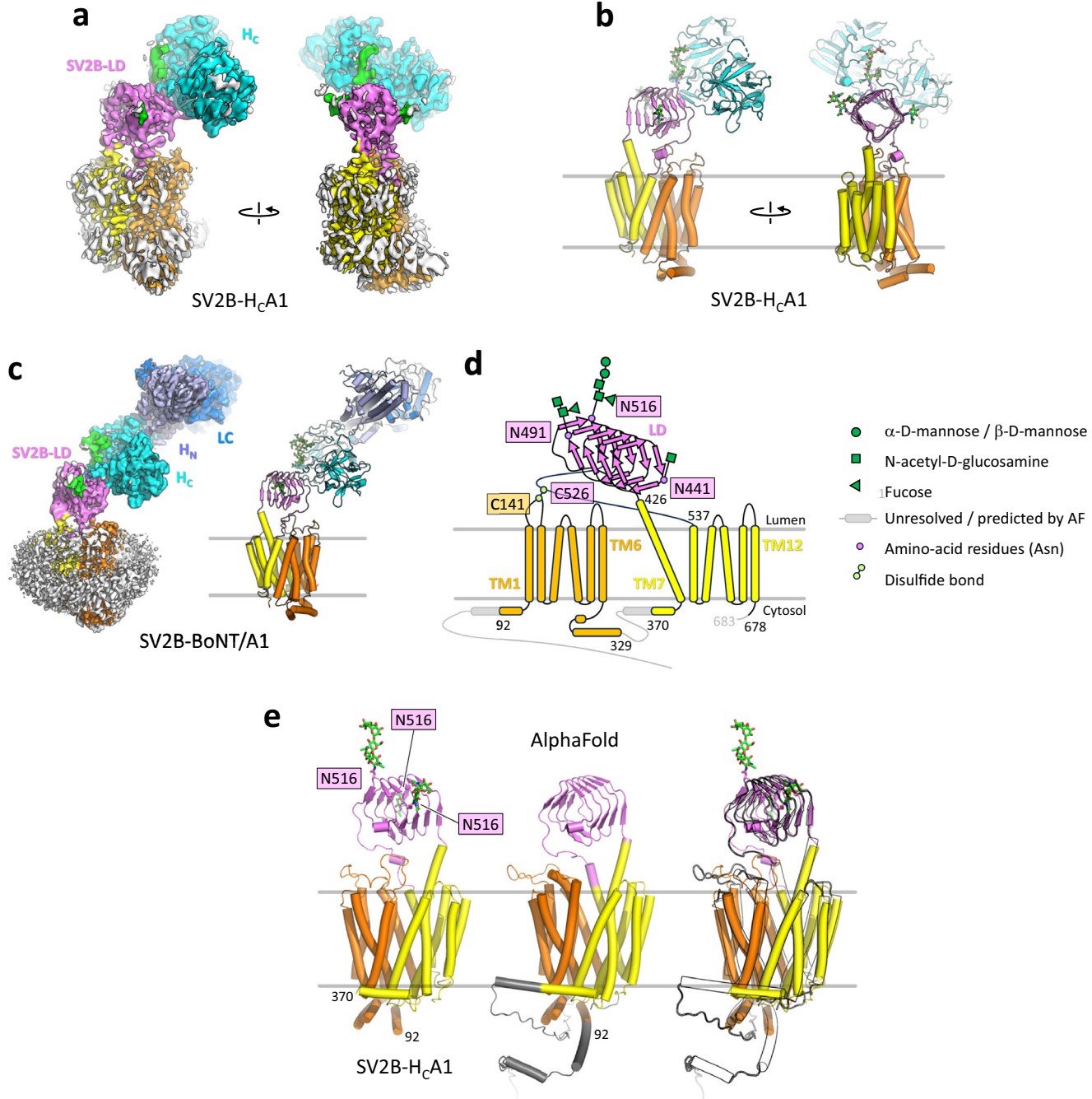

**Fig. 3 | Cryo-EM structures of the SV2B-H_CA1 and SV2B-BoNT/A1 complexes.** Cryo-EM map (**a**) and model (**b**) of the SV2B-H_CA1 complex at 3.7 Å resolution. **c** Cryo-EM map and model of the SV2B-BoNT/A1 complex at 4 Å resolution. **d** A schematic representation of the SV2B structure. Regions colored gray that are not resolved in our cryo-EM map. N-linked glycans, green; LD domain, pink; TM1-6, orange; TM7-12, yellow. **e** Comparison of the cryo-EM- and the AlphaFold-generated model of SV2B based on the SV2B-H_CA1complex.

structure is closed, with the entrance to the putative solute translocation pathway sealed by LL1. Notably, the disulfide bridge between C141 and C526 that potentially allows the LD to act as a lever, controlling the opening or closure of the translocation pathway. Alternatively, transporter activity and the associated conformational rearrangements in LL1 may be conveyed to the LD via the covalent connection, raising intriguing new questions for future investigations.

### BoNT/A1 binds to SV2B in an open state

The conformation of the full-length toxin is important for translocation as well as toxin action, but it is unknown if the conformation

BoNT/A1 in the receptor-bound form would be the same as in the unbound state and how it would relate to its ability to mediate the LC translocation across the membrane[1,61]. As we have seen in our cryo-EM reconstruction of BoNT/A1 in solution, the toxin adopts a predominantly semi-closed conformation (Fig. 2a–c, f). Unexpectedly, BoNT/A1 binding to SV2B resulted in a switch from the semi-closed to the open conformation, the domain arrangement that is observed in the crystal structure of the isolated, full-length toxin (Fig. 3a–c). In this conformation, there is no contact between LC and H_C, which are fully separated by H_N that wraps around LC as the belt and then folds into the elongated helical translocation domain[21–23]. Notably, based on our

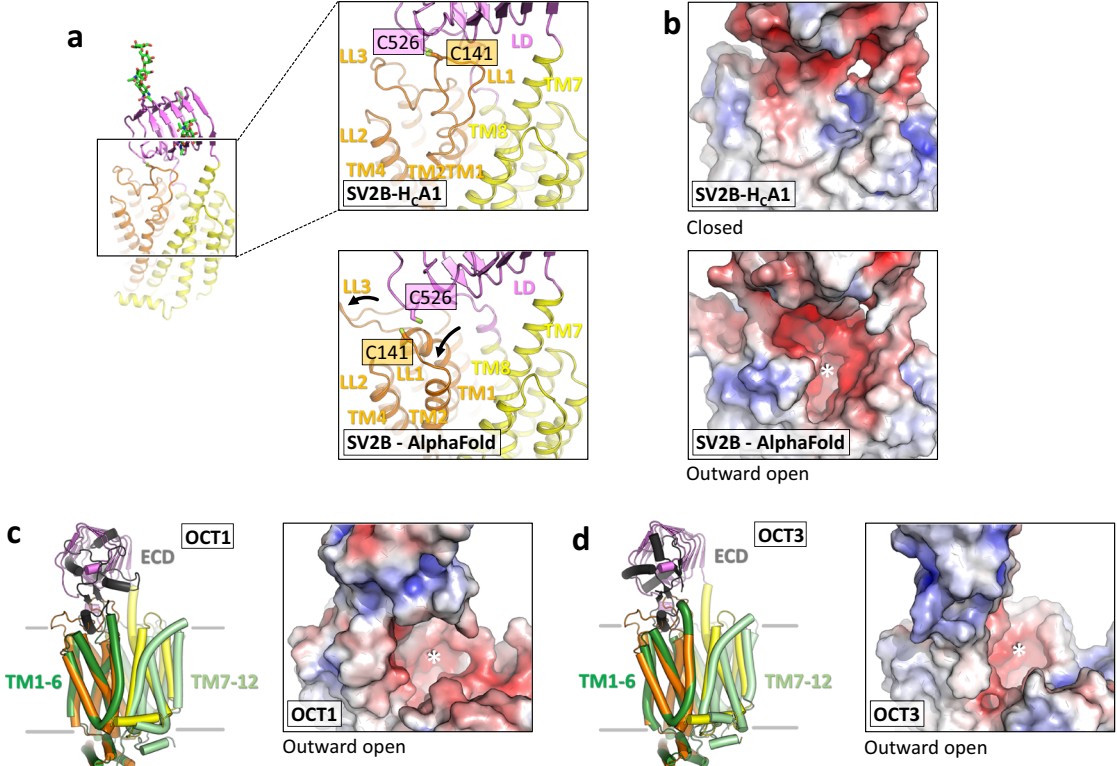

**Fig. 4 | Cryo-EM structure of SV2B. a** Zoom-in view of the LL1-LL4 region of SV2B in the cryo-EM-based and AlphaFold-based model (AF-Q7L1I2-F1). **b** Views in f shown in surface representation colored according to electrostatic potential. **c, d** Cryo-EM structures of OCT1 (PDB8ET6) and OCT3 (PDB7ZH0) determined in outward-open conformations. The same views as in (**b**) were used. Asterisks in indicate the opening of a putative translocation pathway in SV2B (**b**) and show the translocation pathways of Oct1 (**c**) and Oct3 (**d**).

structure, in the open conformation, the $H_N$ and the LC are positioned far away from the plasma membrane or the synaptic vesicle membrane (~40 Å from the lipid bilayer margin; Supplementary Fig. 11A), suggesting that this conformation prevents BoNT/A1 translocation. During intoxication, this state would correspond to that of a receptor-bound toxin on the cell-surface or after uptake into synaptic vesicle prior to acidification.

### The carbohydrate-protein binding interface between BoNT/A1 and SV2B

Glycosylation plays a key role in the interaction between BoNT/A1 and its receptors. Until now the only insights into a glycosylated receptor-toxin interaction could be derived from the structure of a complex formed by the glycosylated SV2C-LD (gSV2C-LD) and the $H_C$A1 (Fig. 6a, b)[30]. For the analysis of the SV2B-BoNT/A1 binding interface we used the SV2B-$H_C$A1 complex because of its higher resolution. Superimposition of the SV2B-$H_C$A1 and gSV2C-LD-$H_C$A1 structures revealed similar protein-carbohydrate interactions (Fig. 6a, b). Both structures share a complex-type N-glycan that is linked to Asn-516 with two N-acetylglucosamines (NAG), two mannoses (BMA) and a fucose (FUC). An additional BMA is visible in our structure in proximity of $H_C$A1, which may be the reason that carbohydrate-protein interactions contribute more to the interactions of BoNT/A1 with SV2B than with SV2C. Notably, these interactions are different in the SV2Ca-LD-$H_C$E1 complex[50] (Fig. 6c).

There are two other putative N-glycosylation sites in SV2B-LD4 that are conserved amongst SV2 family members. Only one NAG and two NAGs and one FUC could be revealed for the N-glycans attached to Asn-441 and Asn-491, respectively, which likely the result of the overall flexibility of the complex glycans at these positions (Figs. 5b, e

and 6a; Supplementary Fig. 8). Both N-glycans are located too far away from the toxin/receptor interface to directly participate in BoNT/A1 binding.

The most prominent feature of the direct interaction of the glycan at N516 and $H_C$A1 is a π-CH stacking interaction between F953 and the sugar ring of the first NAG. The water molecules previously seen at the toxin/receptor interface that further strengthen the glycan-$H_C$A1 interactions[30] are not visible at this resolution. Together, these interactions cover a contact area of 365 Å$^2$.

### The protein-protein binding interface of BoNT/A1 and SV2B

All of the available $H_C$A structures complexed to SV2 reveal high plasticity at the toxin-receptor binding interface[30,47–49]. Rather than a network of conserved side chain-side chain interactions, shape complementarity of the toxin and the receptor at the binding interface appears to be more important for specificity. At the interface 12 SV2B amino-acid residues and 13 BoNT/A1 amino-acid residues are partially or fully buried (Fig. 5c). Together, protein-protein interactions account for a contact area of 619 Å$^2$. Several of these buried residues are engaged in backbone-backbone hydrogen bonds between interacting β-strands (Fig. 5d). Fewer specific side chain-side chain interactions are seen in the structure compared to $H_C$A-SV2C complexes, which most likely reflects the larger contribution of carbohydrate-protein interactions for toxin-receptor binding. Among these, the most prominent is a network of salt bridges and hydrogen bonds involving E521 of SV2B and Y1122, K1137, and R1156 of BoNT/A1 (Fig. 5d). Notably, E521 is one of four residues that are not conserved in the very similar binding region of human SV2A, where the corresponding residue is H578 (Supplementary Fig. 12). This finding suggests that E521 of SV2B not only

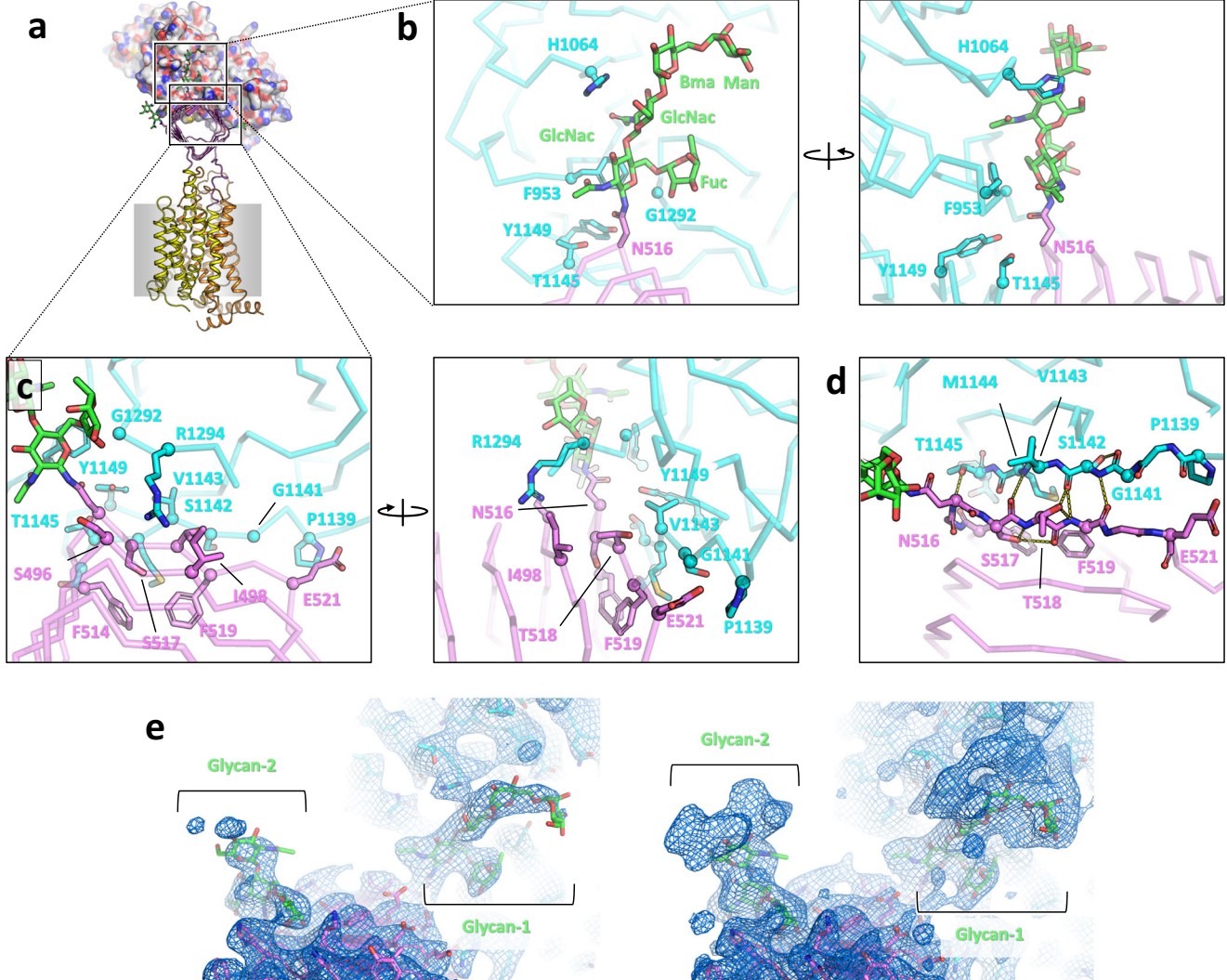

**Fig. 5 | The SV2B-BoNT/A1 binding interface. a** Overall view of the SV2B-H$_C$A1 cryo-EM structure (the H$_C$A1 is shown in surface representation, colored by residue type). Boxed areas corresponding to the close-up views of the carbohydrate-protein (**b**) and protein-protein interactions between SV2B-LD and the Hc of BoNT/A1 (**c, d**). H$_C$A1 and SV2B-LD are shown in ribbon representation; amino-acid residues interacting at the interface are shown as sticks; key hydrogen bonding interactions at the protein-protein interface are shown as yellow dotted lines in (**d**). **e** Views of the carbohydrate cryo-EM densities at two different σ levels (left, 8σ; right, 5σ) indicate the presence of less well-ordered sugar chains.

plays a role in differential SV2 isoform binding by the toxin but also in the variations of interactions observed for BoNT/A subtypes, which seem to be important for differences in toxin function even within the same serotype. Consistent with this conclusion, the corresponding residue in SV2C is also a His (H564). This amino-acid residue is involved in interactions with BoNT/A2 that are not observed in BoNT/A1[48].

### High-affinity toxin-receptor binding is mediated by protein-protein and protein-glycan interactions

Microscale thermophoresis (MST)-based experiments revealed a high-affinity interaction of SV2B with BoNT/A1 and H$_C$A1, with very similar K$_D$ values of 110 nM and 180 nM, respectively (Fig. 6d). These values are similar to the K$_D$ values reported for the interaction H$_C$A1 and gSV2C-LD[30].

The importance of the prominent π-CH stacking interaction between F953 of BoNT/A1 and the first NAG at N516 of SV2B was tested by site-directed mutagenesis. To evaluate the effects on protein-protein interactions, we decided to mutate conserved rather than unique interactions. For this purpose, we mutated F953 to Gly. This approach is based on the superimposition of the SV2B-H$_C$A1 structure

to the previously published H$_C$A-SV2-LD structures[30,47-49], which revealed that the overall binding mode is conserved. Because different side chain-side chain interactions are known to be tolerated at the interface as a result of the high plasticity of the toxin-receptor interaction[48], a single amino acid-residue substitutions might not result in a complete loss of binding of SV2B to BoNT/A1. To avoid this, we used the double BoNT/A1 mutant T1145A-T1146A (TT-AA) which, based on our previous studies with SV2C[47], is expected to disrupt the interaction with the protein moiety of SV2B without affecting glycan binding. These two threonine residues, located at the core of the toxin-receptor protein-protein interface, are conserved in all BoNT/A subtypes[26].

Receptor binding of the mutant toxins was assessed by pull-down assays using H$_C$A1 fused to GST and full-length SV2B (Supplementary Fig. 1F–I). SDS-PAGE analysis demonstrated that both the F953G and the TT/AA mutant of BoNT/A1 completely abrogated binding to SV2B (Fig. 6e), indicating that a combination of glycan-protein and protein-protein interactions is required for the binding of the toxin to the receptor.

Taken together, these findings highlight the conserved binding mode used by BoNT/A1 to recognize the SV2 proteins, which may

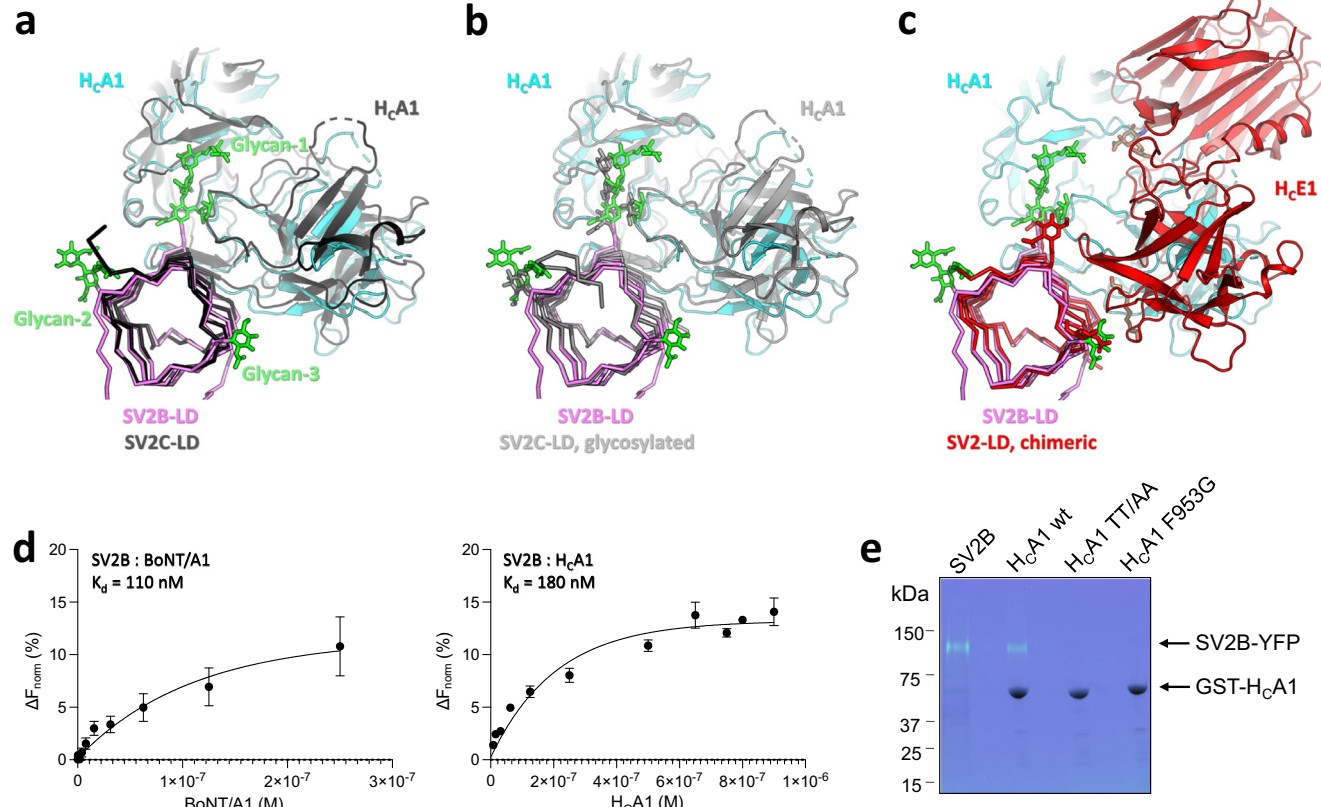

**Fig. 6 | High-affinity binding of BoNT/A1 to SV2B and comparison to other toxin-receptor complex structures. a–c** Comparison of the SV2B-H$_C$A1 cryo-EM structure with the SV2C-LD-H$_C$A1 (f, PDB4JRA, black), gSV2C-LD-H$_C$A1 (g, PDB5JLV, grey) and SV2Ac-LD-H$_C$E1 (h, PDB7UIB) crystal structures. **d** Microscale thermophoresis-based binding assays reveal similar high-affinity interactions between SV2B-LD and H$_C$A1 or BoNT/A1. The results shown are mean values ± SD of six and five independent measurements for SV2B-LD-H$_C$A1 and SV2B-LD-BoNT/A1,

respectively. **e** GST-Sepharose-based pull-down assays using wild-type (positive control) and mutant GST-tagged H$_C$A1 and SV2B. Both H$_C$A1 mutants, F953G and T1145A/T1146A (TT/AA), did not bind to SV2B, demonstrating the importance of both carbohydrate-protein and protein-protein interactions for the binding of the toxin to the receptor. Experiments were repeated three times independently and yielded similar results.

potentially be exploited to fine-tune the medical properties of this therapeutic protein.

## Receptor-bound BoNT/A1 undergoes a pH-dependent switch into a translocation-competent conformation

Conformational changes in BoNTs, particularly in H$_N$, have been proposed to occur upon acidification of synaptic vesicles, resulting in the insertion of the H$_N$ domain into the synaptic vesicle membrane and its conversion into a membrane channel[1,31]. Low pH is thought to initiate these conformational rearrangements. Therefore, we determined the structure of the SV2B-BoNT/A1 complex at pH 5.5 (Supplementary Figs. 3E, 13–16). At low pH, the receptor-bound toxin undergoes radical remodeling, adopting a semi-closed conformation (Fig. 7). The key element of the BoNT/A1 structure mediating the domain reorientation from the open to the semi-closed state is the helical linker region at the boundary between H$_C$ and H$_N$ (Fig. 8a). Reorientation of this linker results in a large-scale conformational change, which brings the LC into close proximity to the synaptic vesicle membrane. The SV2B-BoNT/A1 structure at low pH illustrates that sequences spanning amino-acid residues 2–26, 135–145, 296–315, 323–333, and 513–526 are positioned very close to the micelle/lipid bilayer margin (Fig. 7c and Supplementary Fig. 11B). Furthermore, in this conformation, one of the "edges" of the helical part of H$_N$ is positioned closer to the to the membrane plane (residues I685-L690, Y824-K840), indicative of possible H$_N$-membrane interactions associated with the next stage of toxin translocation (Supplementary Fig. 11). Our findings are consistent with the identification of several elements in BoNT/A1 that have been

implicated in the interactions with the intracellular neuronal plasma membrane, including the membrane localization domain (MLD, residues 275–334) and the N-terminus of LC/A1[62]. This interaction additionally involves the belt of H$_N$, which plays important roles in regulating the rate of toxin translocation[63] and in regulating the interactions between the LC domain and the lipid bilayer[64].

This conformation is very similar to the free toxin conformation in solution at physiological pH (Fig. 2 and Supplementary Fig. 15). Comparison to BoNT/A1 in complex with NTNHA[54] revealed, however, that the H$_C$ domain orientation is completely different, owing to the helical linker reorientation (Figs. 7d and 8a).

It has been suggested that a distinct region of H$_N$, termed the BoNT switch, flips out upon acidification of the synaptic vesical lumen to enable H$_N$ insertion[32]. Notably, there are no obvious conformational changes detectable in the switch region of BoNT/A1 bound to SV2B at low pH compared to the unbound toxin at neutral pH (Fig. 8b). Moreover, the BoNT switch is located in a H$_N$ region opposite and far away from the synaptic membrane. However, the switch region is remarkably close to the helical linker region, suggesting that in the early steps of translocation, the switch may play a role in the pH-dependent conformational change of the toxin, rather than in direct interactions with the lipid bilayer.

## Discussion
The availability of structures of full-length BoNT and SV2 proteins and their complexes is crucial for understanding toxin-receptor recognition and toxin translocation at the molecular detail. Towards this aim,

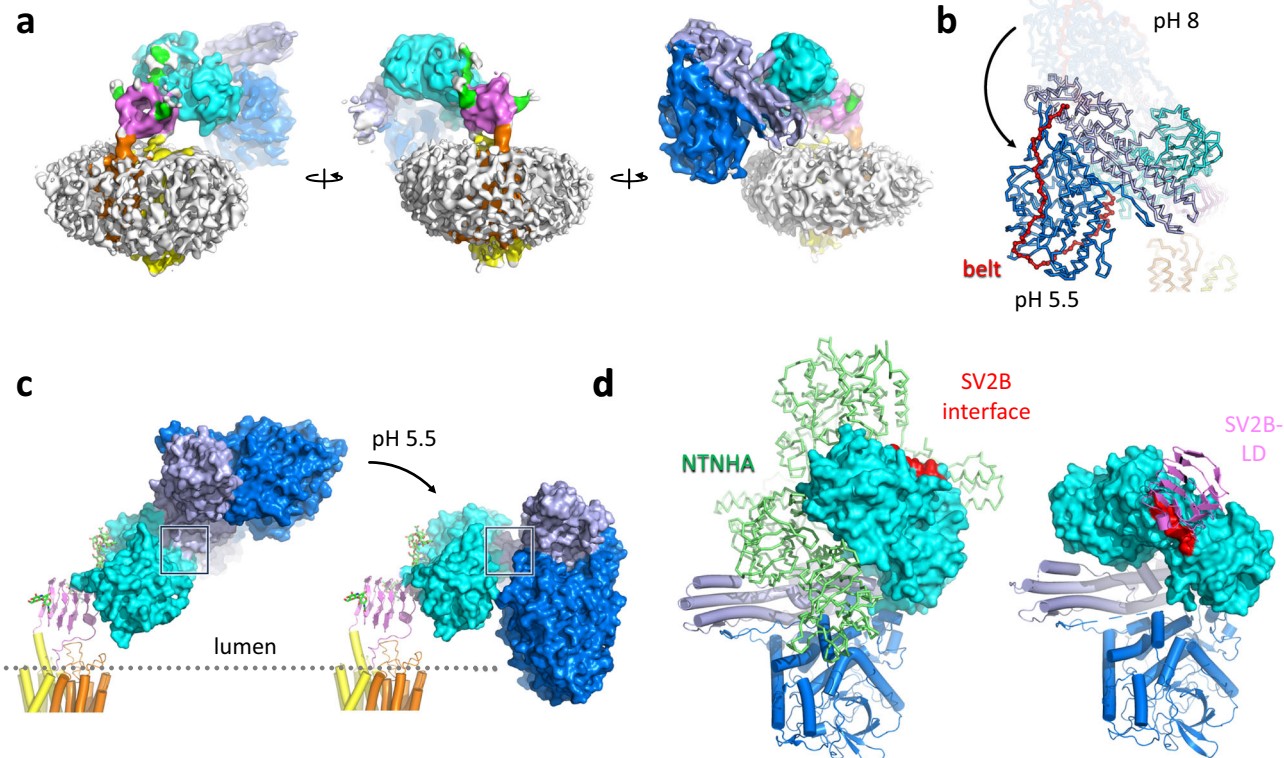

**Fig. 7 | Structure of SV2B-BoNT/A1 complex at pH 5.5. a** Views of the cryo-EM density map reconstructed using the sample prepared at pH5.5. **b**, **c** Comparison of the domain rearrangements of BoNT/A1 from the open to the semi-closed state. The square indicates the region of conformational rearrangements. **d** Comparison of the NTNHA-bound BoNT/A1 (left, PDB3V0B) with the SV2B-bound toxin at pH 5.5.

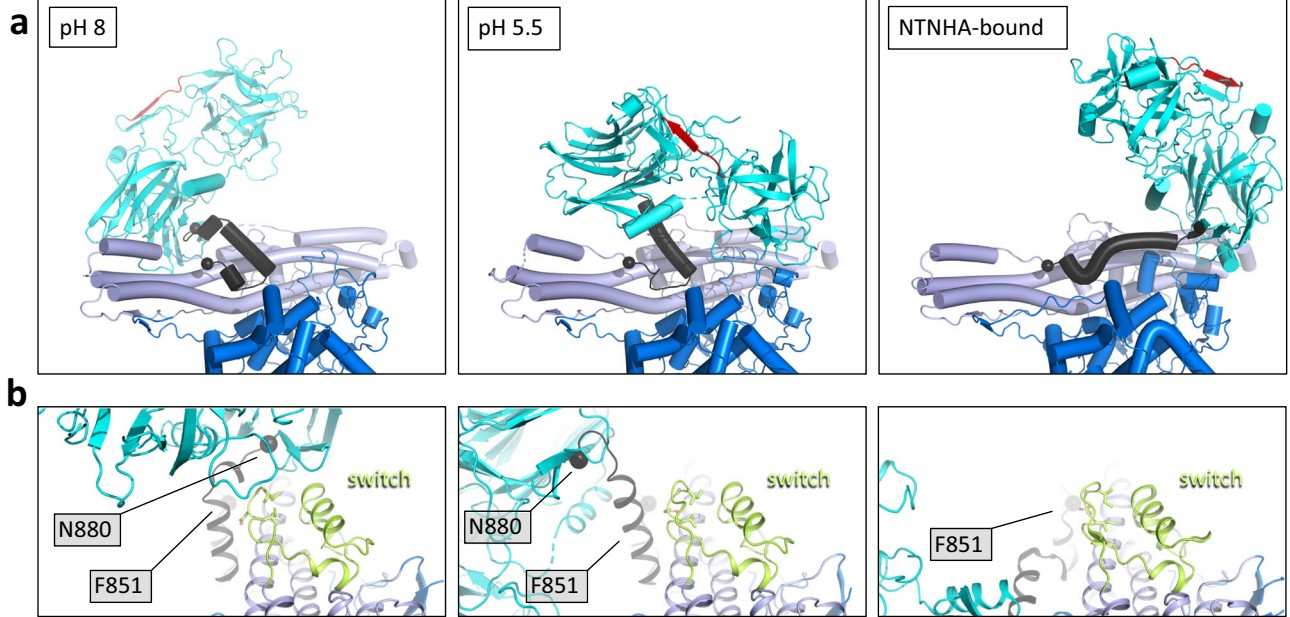

**Fig. 8 | Structural elements important for BoNT/A1 conformation. a** Close-up views of the helical linker region (F851-N880, black) at the interface between $H_C$ and $H_N$ in three structures. Red β-sheets indicate the position of the SV2B-LD interface within $H_CA1$. **b** A view of the linker region in relation to the switch (E820-F667; lime-colored) in the three indicated structures.

we determined the structure of SV2B. SV2B shows the highest structural similarity to organic cation transporters (OCTs), sharing a similar domain architecture.

The LD domain of SV2 proteins has no structural similarity with the extracellular domain (ECD) of OCTs. The ECD of OCT1-3 is inserted in the loop between TM1 and TM2, whereas the LD of SV2 proteins is connected to TM7 and TM8. The LD of SV2 proteins and the ECD of OCTs contain the N-glycosylation sites of the proteins. A very interesting analogy between SV2 and OCT proteins are their partially overlapping expression patterns. While OCTs are expressed mainly in

the kidney and liver, OCT2 is also broadly expressed in the central nervous system where it is preferentially enriched as a potential choline transporter on synaptic vesicles in cholinergic neurons[65]. As revealed by immunoelectron microscopy, OCT2 localizes to synaptic vesicles in presynaptic terminals around the motoneurons and to synaptic vesicles in nerve terminals in NMJs[65]. Notably, cholinergic neurons represent the main target of BoNT/A1. A possible functional significance of the potential co-localization of OCT2 and SV2 proteins for BoNT intoxication remains to be elucidated.

A very exciting aspect of SV2 structure and function is provided by the two states of SV2B, the cryo-EM-based "closed" state, and the AlphaFold-predicted "outward open" state. The strongly negatively charged translocation pathway of SV2B in the open state parallels observations with the OCTs, which transport cationic compounds across the membrane[58,59]. It is therefore tempting to speculate that SV2B, if viewed as an orphan transporter, may also be involved in the translocation of small organic cations or molecules similar to those that can be transported by OCTs. In view of the proton gradient across the synaptic-vesicle membrane, the presence of a negatively-charged translocation channel raises the question whether SV2 proteins could exert their yet unknown functions as transporters coupled to the H+ gradient.

The direct physical link between the LD and the LL1 via a conserved disulfide bond raises a hypothesis about the connection between the SV2 translocation pathway and a putative transport activity, the conformation of SV2 proteins, and the interactions of SV2 with BoNTs. Whole domain conformational rearrangements of LD/LL1 induced by BoNT/A1 (or H$_C$A1) binding may lead to closure of the putative solute translocation pathway, which may be important for toxin translocation. It is also possible that a closed state of SV2 is a prerequisite for efficient BoNT interaction. Although these exciting possibilities are purely speculative at this point, they hint at a previously unappreciated link between a potential SV2 transporter activity and the mechanism of toxin recognition/translocation.

The most significant findings of our work are the conformational changes observed for BoNT/A1. In the absence of its receptor, the toxin adopts a unique, semi-closed conformation that resembles the conformation of TeNT observed at slightly acidic conditions (Fig. 2). It is also reminiscent but distinct from that previously observed for the toxin in the M-PTC with NTNHA[54]. The semi-closed conformation is markedly different from the open conformation seen in the crystal structure of BoNT/A1 or BoNT/B. Upon binding to SV2B, BoNT/A1 switches its semi-closed conformation to the open linear arrangement /Fig. 3). Another dramatic conformational change is observed at pH 5.5, the pH value characteristic of acidified synaptic vesicles, at which the SV2B-bound BoNT/A1 switches back to a semi-closed conformation (Fig. 7),

The distinct observed conformations of H$_C$A1 in the free, NTNHA-bound, and SV2B-bound states at different pH values provide an insight into the remarkable plasticity of the neurotoxin, mediated by the helical linker at the H$_C$-H$_N$ interface (Fig. 8). It is possible that the primary role of the H$_N$ regions previously suggested to be critical for translocation, such as the switch region (residues 620–667)[32], is to influence the conformation of the helical linker. Indeed, the residues 625–631 of the switch region are in close proximity of the helical linker, in the receptor-free and receptor-bound forms of BoNT/A1. In contrast, in NTNHA-bound form the switch is dislodged from the helical linker.

The structure of the receptor-toxin complex revealed that BoNT/A1 binding to SV2B at acidic pH brings the LC in close proximity to the membrane (Fig. 9 and Supplementary Fig. 11). Therefore, the LC appears to be another critical but as yet neglected factor contributing to the onset of toxin translocation. Consistent with our findings, LC/A1 was previously shown to localize to the intracellular neuronal plasma membrane[66]. The stable association of LC/A1 with the membrane is mediated by the MLD (residues 275–334) and the N terminus of the

toxin[62]. In agreement with these findings, in our low-pH cryo-EM structure of the receptor-toxin complex, the N terminus (P2 to V17) and two loops within the MLD are juxtaposed with the synaptic vesicle membrane, likely contributing directly to the protein-lipid interaction (Fig. 9 and Supplementary Fig. 11). The overall charge of the LC elements interacting with the membrane is positive. The involvement of the belt in synaptic vesicle membrane binding seen in our low-pH toxin-receptor structure raises the question about the functional relevance of this interaction. A possible answer is provided by the suggestion that the belt acts as regulator of membrane interaction[64]. Consistent with this proposal, several negatively-charged amino-acid residues of the belt are found in the interface with the membrane (e.g.: D509, E511, E513, E518, D523). Acidification of the synaptic vesicle lumen might neutralize these residues, abrogate electrostatic repulsion to negatively-charged lipid head-groups and allow LC to interact with the membrane, suggesting a role of the belt in toxin-translocation specificity.

Our conclusions are consistent with the results of a mutational study of three conserved negatively-charged amino acids in BoNT/B (E48 and E653 to Q; D877 to N) whose side chains are predicted to be protonated at physiological synaptic vesicular pH[67]. The triple mutant showed increased neurotoxicity as a result of faster LC translocation, indicating that neutralization of specific negative surface charges facilitates the interaction of the toxin with the synaptic vesicle membrane.

It is noteworthy to mention that our low-pH cryo-EM structure of the toxin-receptor complex would also be compatible with alternative models of toxin translocation. It has been reported that LC interacts with and can permeabilize anionic lipid bilayers at acidic pH without the assistance of H$_N$ and without changing its secondary structure[68]. The functional role of H$_N$ in translocation is to act as a chaperone for LC stabilization and as a mediator of its interactions with the membrane. The closed conformation of SV2B-bound BoNT/A1 also moves one edge of H$_N$ (two amino acid stretches, rich in positively charged residues: I685-L690, Y824-K840) near to the synaptic vesicle membrane, suggesting that this region might be the first point of membrane contact. In contrast, the BoNT/A1 switch[32] is located on the H$_N$ region opposite and far away from the membrane, indicating a possible role of this structural motif in later stages of translocation.

The observation that SV2B binding of BoNT/A1 at neutral pH results in an open conformation of the toxin raises questions about the functional significance of this domain arrangement. Because in the open conformation of the SV2B-bound toxin the LC and H$_N$ are far away from the synaptic vesicle membrane, the role of this conformation appears to be the prevention of the interaction of the two domains with membranes and premature translocation before the toxin/receptor complex reaches its proper destination. Consistent with this hypothesis, it has been demonstrated that during neuronal cell binding and receptor-mediated endocytosis H$_C$A prevents transmembrane channel formation of H$_N$ until acidification within synaptic vesicles[69]. Accordingly, removal of H$_C$ resulted in spontaneous channel formation that was independent of pH. Notably, BoNT/A devoid of H$_C$ can enter neurons probably through the cell membrane and cleave SNAP25 at an efficiency that was lower than the full-length toxin. Therefore, the H$_C$ and the conformation of the receptor-bound toxin appear to be crucial for its translocation specificity.

The conformations of full-length BoNTs have been implicated in the speed of LC translocation as well as onset of toxin action[22]. Serotypes BoNT/E and BoNT/F have a shorter onset and duration of action compared to BoNT/A1 and therefore are potentially very attractive toxins for applications in certain therapeutic areas[70]. The closed conformation characteristic of BoNT/E is also predicted for BoNT/F by AlphaFold. It has been suggested that the unique domain arrangement of BoNT/E is the likely reason for its faster translocation and onset of action[22]. Based on the unique conformation of BoNT/E, it has also been speculated that for successful translocation the open form of BoNT/A and B will need to convert into a closed, translocation-competent

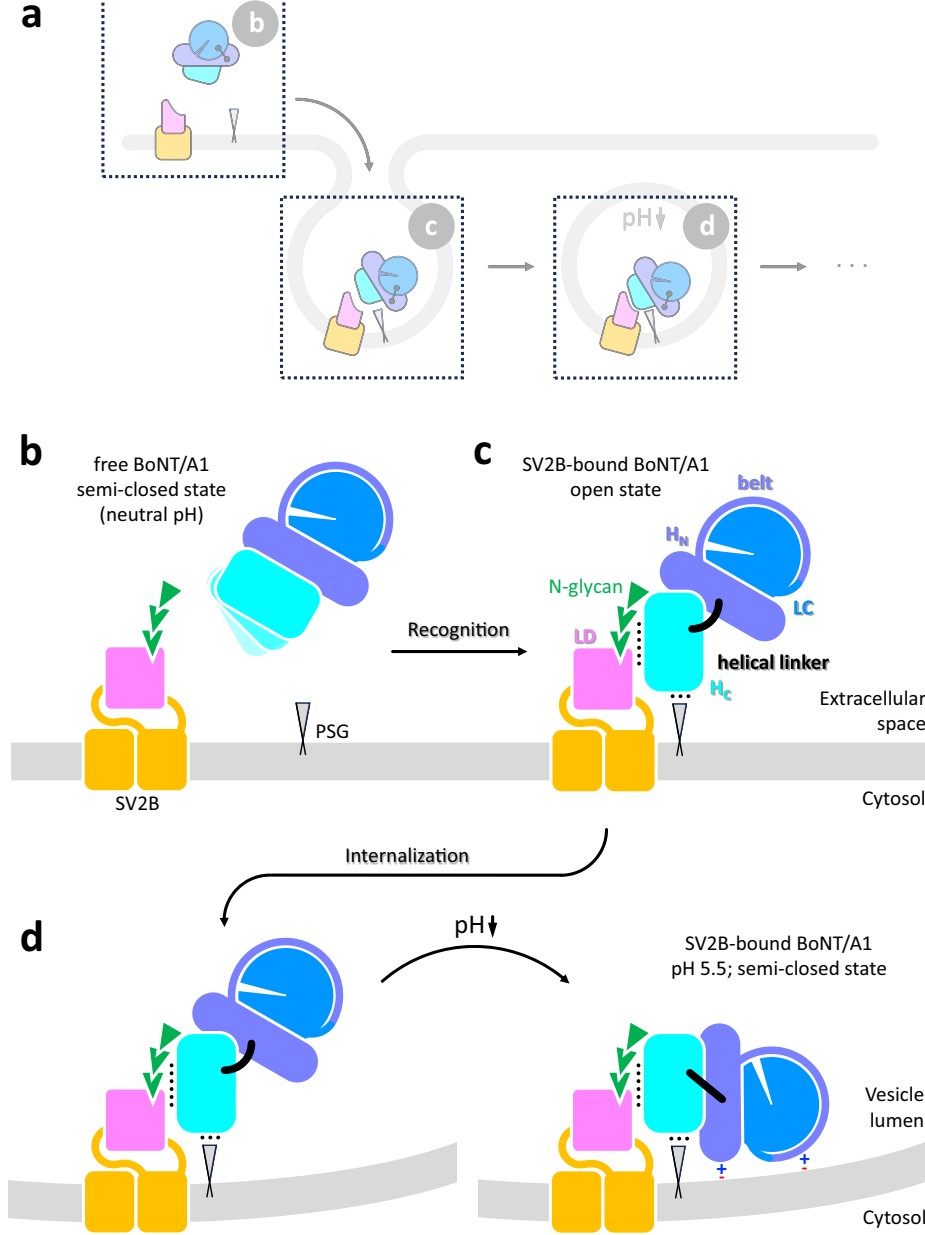

**Fig. 9 | Proposed model of the initial stages of BoNT/A1 translocation. a** Early stages of BoNT/A1 recognition (as in Fig. 1). **b** In solution, the neurotoxin adopts a dynamic semi-closed conformation. **c** Upon receptor binding at the physiological extracellular pH, BoNT/A1 adopts an open conformation that prevents the interaction of LC and $H_N$ with the neuronal cell membrane and the synaptic vesicle membrane. **d** Acidification of the vesicle leads to a dramatic conformational change in BoNT/A1 to a semi-closed state. Rearrangement of the helical linker at the $H_C$-$H_N$ boundary is key for the conformational change of the toxin. The domain rearrangement brings the LC, the belt region of $H_N$ that contains several negatively-charged residues that are potentially neutralized at low pH, and one of the edges of $H_N$ that is rich in positively charged residues in close proximity to the synaptic vesicle membrane. This conformational change appears to prime the toxin for the subsequent translocation into the neuronal cytosol.

conformation like the one seen in BoNT/E. This hypothesis is fully consistent with our findings. Cryo-EM studies on TeNT support our findings on a pH-dependent conformational switch from the open into the semi-closed form of SV2B-bound BoNT/A1[53]. In these studies, the open conformation of isolated TeNT could be switched from an open into a compact form via a semi-open intermediate upon acidification.

The dramatic conformation differences between BoNT/A1 alone and the SV2B-bound form of the toxin have important implications for translocation. Further work is required, however, to fully resolve the relationship between toxin conformation and translocation efficiency. Additional high-resolution structures of full-length BoNTs bound to SV2 will be needed to capture different stages of toxin translocation. The structures of the SV2B-BoNT/A1 complex described in this study will provide a solid basis for obtaining these additional toxin-receptor complex structures. They will further contribute towards our understanding of toxin action, which is crucial for the design of improved BoNTs for medical applications.

## Methods
### Reagents and chemicals
Detergents, dodecyl-β-maltoside (DDM), glyco-diosgenin (GDN), and cholesteryl hemisuccinate (CHS) were purchased from Anatrace Inc.

All other chemicals were obtained from Sigma-Aldrich (St. Louis, MO, USA) unless indicated otherwise.

## Constructs

Inactive full-length BoNT/A1 (Uniprot ID: P0DPI1) in pQE30 was generously provided by Dr. Thomas Binz, (Hannover Medical School MHH, Hannover, Germany). BoNT/A1 contains three inactivating mutations, E224Q, R363A, and Y366F, and carries an N-terminal 6xHis tag and a C-terminal Twin-Strep-tag. The DNA constructs encoding the full-length human SV2A (Uniprot ID: Q7L0J3), SV2B (Uniprot ID: Q7L1I2), and SV2C (Uniprot ID: Q496J9) proteins were synthesized, and cloned by Genewiz into a pACMV-based vector with a C-terminal 3C-YFP-Twin-Strep fusion tag. Wild-type $H_CA1$ (UniProt entry P0DPI1) encoding amino acids 871–1296 was amplified by PCR using a cDNA template that was codon-optimized for expression in *E. coli* (Genewiz). The primer sequences used were: $H_CA1$ F, 5′- CTC GTC GGA TCC AAG TAC ACC ATG TTC CAC TAT CTG-3′ and $H_CA1$ R, 5′-CTC GTC GAA TTC TCA TTA CAG CGG ACG TTC ACC C-3′. The insert was cloned into a modified pET-22b vector containing an N-terminal 6×His. For the generation of the $H_CA1$ mutants T1145A/T1146A and R593G, the insert was cloned into a pET-based vector supplying a 6×His- GST fusion tag. Mutants were prepared by using a modified Quikchange protocol and primer sets $H_CA1$ F953G F, 5′-CCT AAA TAC GGC AAC TCC ATC TCC CTC AAC AAC GAG TAT ACG-3′ and $H_CA1$ F953G R, 5′-GAT GGA GTT GCC GTA TTT AGG GAT GCG AAT CCA AAA AGA GGT GC-3′, and $H_CA1$ TTAA F, 5′-CTG TTA TGG CCG CGA ACA TTT ACC TCA ATT CCT CCC TGT ACC G-3′ and $H_CA1$ TTAA R, 5′-GTT CGC GGC CAT AAC AGA GCC ACG CGG ACC C-3′. All insert sequences were verified by DNA sequencing (Eurofins GATC).

## Protein expression and purification

**BoNT/A1.** Inactive full-length BoNT/A1 protein was expressed in *E. coli* strain BL21(DE3) (New England Biolabs). Bacteria were cultured at 37 °C in TB medium containing 100 μg/ml Amp for selection until an $OD_{600}$ of 1.5 was reached. The temperature was then lowered to 30 °C, expression was induced with 0.5 mM IPTG, and incubation continued at 30 °C for -16 h. The cells were harvested by centrifugation (4000 × $g$, 4 °C, 15 min) and stored at −80 °C until further use. For purification, cell pellets corresponding to 4 L culture were thawed and resuspended in freshly prepared lysis buffer (100 mM Tris-HCl pH 8.0, 150 mM NaCl, supplemented with cOmplete™ EDTA-free protease inhibitor cocktail (Roche), DNAse, lysozyme, 10 mM $MgCl_2$, 1 mM PMSF) and lysed with a microfluidizer LM-10 (Instrument) at 18 kPsi. The lysate was centrifuged at 24,500 × $g$ for 45 min at 4 °C, before loading onto a 5 ml StrepTRAP HP column (GE Healthcare) pre-equilibrated with phosphate buffered saline buffer (PBS, pH 7.4). After one washing step with 2 column volumes (CV), BoNT/A1 was eluted with PBS buffer containing 5 mM desthiobiothin (IBA Lifesciences). Pooled fractions of eluted protein were subjected to size exclusion chromatography on a Superdex 75 16/600 column (GE Healthcare) pre-equilibrated in PBS buffer. Sample purity and identity were assessed by SDS-PAGE (Bio-Rad) and western blot analysis (anti-Strep, 1:2000; Invitrogen; goat anti-mouse 1:4000; Invitrogen).

**$H_CA1$.** Wild-type and mutant $H_CA1$ domains were expressed in *E. coli* strain BL21(DE3) (New England Biolabs). Precultures (100 ml Luria Bertani (LB) medium supplemented with 100 μg/ml Amp) were grown overnight at 37 °C in a shaking incubator. LB medium (4 l supplemented with 100 μg/ml Amp) was inoculated 1:40 with the overnight culture, and the bacteria were grown at 37 °C until an $OD_{600}$ of 0.9 was reached. Subsequently, the temperature was lowered to 20 °C, expression was induced with 0.5 mM IPTG, and the incubation was continued overnight. The cells were harvested by centrifugation. The cell pellets were resuspended in 100 ml lysis buffer (50 mM Tris, pH 7.5, 500 mM NaCl, 10 mM imidazole, 10 mM β-mercapthoethanol

and 1 cOmplete EDTA-free protease inhibitor cocktail tablet (Roche Diagnostics)). The cells were lysed on ice by ultrasonication. Lysate clearing was performed for 1 h at 24,500 g, and the resultant supernatant was passed through a 0.45 μm filter. The proteins were subsequently purified by IMAC (on a 5 ml HisTrap FF Crude column, GE Healthcare) and gel filtration (HiLoad 16/60 Superdex 200, GE Healthcare). The proteins were subsequently purified by Ni-NTA affinity chromatography (5 ml HisTrap FF column, GE Healthcare) pre-equilibrated with 20 mM Tris, pH 7.5 400 mM NaCl, 10 mM imidazole. After with 5 CV, the proteins were eluted with high imidazole buffer (20 mM Tris, pH 7.5, 400 mM NaCl, 400 mM imidazole). Pooled fractions of eluted protein were subjected to size exclusion chromatography (SEC) (HiLoad 16/60 Superdex 200, GE Healthcare) pre-equilibrated 20 mM Tris-HCl, pH 7.5, 150 mM NaCl.

**GFP nanobody.** The expression and purification of GFP nanobody was carried out as previously described[71]. Anti-GFP nanobody in *E. coli* strain BL21(DE3) (New England Biolabs) was grown in LB medium supplemented with 100 μg/ml ampicillin at 37 °C. Once the $OD_{600}$ reached 0.5, protein expression was induced by adding 0.5 mM IPTG followed by overnight incubation at 20 °C. Bacterial cultures were harvested by centrifugation at 4000 × $g$ for 30 min at 4 °C, and pellets were frozen in liquid N2 and stored at −80 °C until further use. Frozen pellets were thawed on ice, re-suspended in lysis buffer containing 25 mM HEPES pH 8.0, 150 mM NaCl, 10 mM imidazole, 1 mM PMSF, 10 μg/ml DNase I. The cells were lysed with microfluidizer LM-10 (instrumat) at 18 kPsi, and the cell lysate was centrifuged for 45 min at 24,500 × $g$ at 4 °C. The clarified lysate was loaded onto 5 ml Ni-NTA affinity chromatography column (GE Healthcare), washed with 20 CV of buffer containing 25 mM HEPES pH 8.0, 150 mM NaCl, 20 mM imidazole and eluted with 250 mM imidazole. The eluted protein fraction was concentrated with a 10 kDa cut-off Amicon concentrator (Millipore) and subjected to SEC on a Superdex-75 16/600 GL column (GE Healthcare) in a buffer containing 20 mM HEPES, pH 7.5, and 150 mM NaCl. SEC fractions corresponding to GFP-nanobody were pooled and flash frozen in liquid $N_2$ and stored at −80 °C.

**Human SV2 proteins.** The plasmid for SV2 expression (SV2A, B, or C) was transfected using polyethyleneimine (PEI) into HEK293F cells (Thermo Fisher Scientific) grown to a density of $1.8 \times 10^6$/ml in Gibco® FreeStyle™ 293 Expression Medium (ThermoFisher Scientific). After 48 h, the cells were harvested by centrifugation (800 × $g$ for 20 min at 4 °C), cell pellets were frozen and stored at −80 °C. On the day of purification, cell pellets were thawed and resuspended in 50 mM Tris-HCl pH 8.0, 150 mM NaCl, 10 % glycerol supplemented with protease inhibitors (1 mM benzamidine, 1 μg/ml leupeptin, 1 μg/ml aprotinin, 1 μg/ml pepstatin, 1 μg/ml trypsin inhibitor and 1 mM PMSF). The cells were lysed using a Dounce homogenizer and the total cell membranes were collected by ultracentrifugation (Ti45 rotor, 186,000 x g for 40 min at 4 °C). The membranes were resuspended in a buffer (50 mM Tris-HCl pH 8.0, 150 mM NaCl, 10 % glycerol) containing 1 % DDM and 0.02 % cholesteryl hemisuccinate (CHS) and incubated for membrane protein solubilization at 4 °C for 1 h with rotation. The lysate was cleared by ultracentrifugation (Ti45 rotor, 186,000 × $g$ for 45 min at 4 °C) to remove insoluble cellular debris. The supernatant was incubated with 4 ml of CNBr-activated Sepharose coupled to an anti-GFP nanobody[71]. After a 60 min incubation at 4 °C the resin was collected in a gravity column and washed with 40 CV of 50 mM Tris-HCl pH 8.0, 150 mM NaCl, 0.02 % GDN. The protein was eluted using cleavage by HRV 3 C protease (1:10 w/w). The eluted protein was concentrated with a 100 kDa cut-off Amicon Ultraspin device (Millipore) and subjected to SEC using a Superose 6 Increase 10/300 GL column (GE Healthcare) equilibrated in 25 mM Tris-HCl pH 8.0 (or 25-mM Na-acetate pH 5.5) supplemented with 150 mM NaCl and 0.02 % GDN. The fractions corresponding to purified SV2 (elution volume, 14.5–16 ml) were

collected, concentrated, and used directly for cryo-EM grid preparation. The protein purity was assessed using 4–20% SDS-PAGE (BIO-RAD) and visualized by standard Coomassie brilliant blue staining.

## Microscale thermophoresis (MST) analysis

MST was performed on a NanoTemper Monolith 2020 (TNG) (MM-068) apparatus (Nano Temper Technologies). The Protein Labeling Kit RED-NHS 2nd (Nano Temper Technologies, Germany) was used to label 10 µM of recombinant BoNT/A1 according to the manufacturer's protocol. Labeled BoNT/A1 (diluted to 20 nM) was incubated with increasing concentrations of hSV2B (0.03–1000 nM) in PBS buffer in a ratio of 1:1. The samples were then loaded into standard glass capillaries (Monolith Capillaries, Nano Temper Technologies), and MST analysis was performed (settings for the light-emitting diode and infrared laser were 40%). The results shown are mean values ± SD of five measurements.

In another set of experiments, recombinant $H_CA1$ (20 nM) was incubated with increasing concentrations of hSV2B (0.03–1000 nM) in PBS buffer in a ratio of 1:1. Labeling procedure and measurement was carried out as described above for BoNT/A1. The results shown are mean values ± SD of six measurements. Raw data treatment was done in MO. Affinity Analysis v3.0.5 control software (NanoTemper Technologies) and curve fitting was done using GraphPad Prism 8.3.1, using the full binding equation for a one-to-one model.

## Pull-down assays

Aliquots of purified wild-type and mutant GST-$H_CA1$ (2 µM each) were centrifuged separately at $16,000 \times g$ for 30 min at 4 °C. The proteins were then incubated with ~40 µl Glutathione Sepharose 4B resin (Cytiva) for 45 min at RT. The GST Sepharose resin was washed five times with pull-down buffer: 50 mM Tris-HCl, pH 7.8, 150 mM NaCl, 0.02% GDN. Solubilized hSV2B membrane pellet was added to the resin and incubated for 30 min at room temperature. After incubation, the resin was washed five times with pull-down buffer and proteins were eluted in 50 µl pull-down buffer supplemented with 10 mM reduced glutathione. The eluted fractions were analyzed by SDS-PAGE (Bio-Rad).

## Cryo-EM sample preparation

Samples of the SV2B-$H_CA1$ complex were prepared by mixing concentrated SV2B with $H_CA1$ in 1:1.2 molar ratio. The SV2B-BoNT/A1 complex was prepared by mixing BoNT/A1 and SV2B in 1:1.2 molar ratio and the SV2B-BoNT/A1-GT1b complex included additional 20 molar excess of ganglioside GT1b. Final protein concentrations were 0.4–0.5 mg/ml for BoNT/A1 (pH 7.4), 4–5 mg/ml for SV2B-$H_CA1$ (pH 8.0), 2.5–3.5 mg/ml for SV2B-BoNT/A1-GT1b (pH 8.0), and 3–4 mg/ml for SV2B-BoNT/A1 (pH 5.5).

All the cryo-EM grids were prepared similarly unless otherwise mentioned. In brief, the cryo-EM grids (Quantifoil R1.2/1.3 200 mesh grids) were freshly glow discharged for 25 s at 30 mA in air using an PELCO easiGlow (Ted Pella) glow discharge cleaning system. A 3.5 µl aliquot of concentrated protein sample was applied on the carbon side of the EM grids. The excess of protein sample was blotted for 3 s (blot force of 20) and then plunged into liquid ethane using a Vitrobot Mark IV (ThermoFisher Scientific) with 100% humidity at 4 °C. The frozen grids were stored in liquid nitrogen.

## Cryo-EM data collection and image processing

The data collection for all samples was performed using EPU on a 300 kV Titan Krios (ThermoFisher Scientific) equipped with a Gatan K3 direct electron detector and a Gatan Quantum-LS GIF at ScopeM, ETH Zurich. All movies were acquired in super-resolution mode as 40 frame movies with a defocus range of −0.6 to −2.4 µm. Movies were saved after two-fold binning with final calibrated pixel size of 0.65 Å.

All movies were assigned to distinct optics groups according to the EPU beam shift values using a script provided by Dr. Pavel

Afanasyev (ETH Zurich; https://github.com/afanasyevp/afis). The subsequent steps of cryo-EM data processing were performed in Relion (v3.1, v4.0 and v5.0) or CryoSPARC v4.4.0.

**BoNT/A1.** The cryo-EM dataset of BoNT/A1 composed of 8361 movies was acquired with a total dose of 80 e/Å². The dose was chosen to enhance the contrast of the small and conformationally dynamic BoNT/A1 particles. The movies were motion corrected in RELION 3.1.3 using MotionCorr 1.4.0[72] and micrographs were contrast-transfer-function-(CTF)-corrected using GCTF[73]. Micrographs with an estimated resolution worse than 4 Å were discarded. The remaining 8014 micrographs were subjected to automated particle picking using the Laplacian-of-Gaussian program in RELION 3.1.3 to obtain initial particle set[74]. A total of 4,517,685 particles were picked from the selected micrographs. After iterative 2D classifications, 921,038 particles were imported to CryoSPARC[75], and ab-initio 3D reconstructions were performed. The initial models were subjected to heterogenous 3D classification. The best class from this job with 294,704 particles was then subjected to homogenous refinement and local refinement, which resulted in a density map at an overall resolution of 3.67 Å. To further improve the quality of the map for $H_C$, 3D classification without alignment was performed after masking the LC. Particles from the two best 3D classes with a better resolved $H_C$ (147,560 particles in total) were merged and used for another round of 3D refinement, yielding a 3.8 Å resolution map for the full toxin. To improve resolution further, 121,176 particles corresponding to the micrographs with a CTF fit better than 3.5 Å were sorted. Finally, local refinement with $H_C$ masked yielded a 3.7 Å resolution map. Local resolution maps were calculated with unmodified half maps using the blocres sub-program implemented in CryoSPARC[76]. The detailed steps of cryo-EM processing are depicted in Supplementary Fig. 2.

**SV2B-$H_CA1$.** Cryo-EM datasets of SV2B-$H_CA1$ were acquired in three sets of 12,699, 6814, and 5789 movies, with total doses of 65 e/Å², 58 e/Å², and 58 e/Å², respectively. The movies were then motion-corrected in RELION's own implementation of the UCSF MotionCor2 program in RELION v4.0[77] and micrographs were CTF-corrected using CTFFIND-4.1[73] Micrographs with an estimated CTF resolution worse than 4 Å were discarded. Initially, around 5.5 million particles were auto-picked from dataset 1 using the Laplacian-of-Gaussian program in RELION v4.0. After multiple rounds of 2D and 3D classification, a 3D class showing good characteristic features of the SV2B-$H_CA1$ complex was then used to repick particles for all three datasets with 3D projections. 2D classification of ~5.7 million auto-picked particles yielded ~1.2 million particles. Then 3D classification was performed with 8 classes, which resulted one distinct class with 275,501 particles that could be refined to 4.33 Å resolution. This refined set of particles was then subjected to several iterative cycles of 3D refinement, CTF refinement and particle polishing, yielding 3.90 Å resolution. To improve resolution further, the refined particles were imported to CryoSPARC[75]. The local refinement of 185,478 particles with a CTF value better than 3.5 Å, yielded the final 3.67 Å resolution map. Local resolution maps were calculated with unmodified half maps using the blocres sub-program implemented in CryoSPARC[75,76]. The detailed steps of cryo-EM processing are depicted in Supplementary Fig. 6.

**SV2B-BoNT/A1 (pH 8.0).** Cryo-EM data sets for SV2B-BoNT/A1 at pH 8.0 were acquired as two sets with 13,915 and 14,542 movies and total doses of 58 e/Å² and 50 e/Å², respectively. The movies were motion corrected using RELION's own implementation program and micrographs were CTF corrected using CTFFIND-4.1[74]. Micrographs with an estimated CTF resolution worse than 4 Å were discarded. Around 8.4 million particles were auto-picked from all the micrographs using the Laplacian-of-Gaussian program in RELION v4.0[77]. After multiple rounds

of 2D classification, a set of 349,508 particles was rescaled, and then imported to CryoSPARC for ab-initio 3D reconstructions. The initial 3D models were subjected to heterogenous 3D refinement. Non-uniform refinement of the best class with 149,718 particles yielded a resolution of 3.98 Å. Iterative focused local refinement and no align 3D classification of SV2B without the transmembrane domain (SV2B-LD-BoNT/A1) yielded a 3.7 Å resolution map. To improve resolution further, particles from micrographs with an estimated CTF resolution worse than 3.5 Å were discarded, and local refinement of the remaining 129,039 particles was performed, yielding a 3.49 Å resolution map for SV2B-LD-BoNT/A1. Local resolution maps were calculated with unmodified half maps using the blocres sub-program implemented in CryoSPARC. The detailed steps of cryo-EM processing are depicted in Supplementary Fig. 7.

**SV2B-BoNT/A1 (pH 5.5).** Cryo-EM datasets for SV2B-BoNT/A1 at pH 5.5 were acquired as two sets with 24,088 and 6052 movies and a total dose of 50 e/Å$^2$ each. The movies were motion corrected using RELION's own implementation of UCSF MotionCor2 program[72,77]. Micrographs were then imported to CryoSPARC[75], and CTF correction was performed using the patch CTF (multi) program. Micrographs with an estimated CTF resolution worse than 4 Å were discarded. Around 8.7 million particles were auto-picked from 29,053 micrographs using the blob-picker program. Around 1,373,653 particles from the 2D classification were used for ab-initio 3D reconstructions and heterogenous 3D refinement. After multiple rounds, a set of 275,421 particles with clear secondary structure features for both SV2B and BoNT/A1 was obtained, which after non-uniform refinement yielded a 5.88 Å resolution map. To improve the map further, another round of 2D classification was performed to remove bad particles. Finally, non-uniform refinement of 187,001 particles, followed by local refinement, gave a 5.41 Å resolution map for the SV2B-BoNT/A1 complex. To improve resolution further, iterative focused local refinement of individual domains or combinations of two domains of the SV2B-BoNT/A1 complex was carried out, yielding better a quality of maps with a resolution ranging from 4.2 to 4.5 Å. The overall resolution of the SV2B-LD-BoNT/A1, SV2B-LD-H$_C$A1, BoNT/A1, and LCH$_N$A1 domain focused maps was 4.39, 3.94, 4.30, and 4.25 Å, respectively. Local resolution maps were calculated with unmodified half maps using the blocres sub-program implemented in CryoSPARC[75,76]. The detailed steps of cryo-EM processing are depicted in Supplementary Fig. 13.

### Model building, refinement and validation

The starting coordinates for model building of the SV2B-H$_C$A1 complex was derived from an AlphaFold model[78] of SV2B (AF-Q7L1I2-F1) and the previously solved X-ray crystal structure of H$_C$A1 (PDB2VU9). The coordinates for LCH$_N$A1 or BoNT/A1 modeling were obtained from a previously solved X-ray crystal structure (PDB6UL6). These initial models were then docked into the cryo-EM maps and subjected to flexible fitting with the "all-atom refine" sub-module in COOT[79], followed by several iterative rounds of manual readjustments of the side chains in COOT. The carbohydrates in SV2B were added using the Glyco module in COOT[79,80]. Finally, refinement of the rebuilt model against the EM map was carried out with phenix.real_space_refine[81]. These refined models of SV2B-H$_C$A1 and BoNT/A1 were subsequently used for generating the SV2B-BoNT/A1 complex models. All the models were validated by the Molprobity program in PHENIX[82]. Data collection, processing, and model refinement statistics are provided in Supplementary Table 1. Because the models of BoNT/A1, SV2B-BoNT/A1 (pH 8), and SV2B-BoNT/A1 (pH 5.5) are based on cryo-EM maps of limited resolution, the precise positions of the side-chain atoms in many instances are not well defined.

### Reporting summary

Further information on research design is available in the Nature Portfolio Reporting Summary linked to this article.

## Data availability

Source data are provided with this paper. The cryo-EM maps have been deposited in the Electron Microscopy Data Bank under accession codes EMD-50139 (BoNT/A1), EMD-50138 (local refinement map of LCH$_N$A1 on EMD-50139), EMD-50135 (SV2B-H$_C$A1), EMD-50147 (SV2B–BoNT/A1), EMD-50146 (local refinement map of SV2B-LD–BoNT/A1 on EMD-50147), EMD-50151 (SV2B–BoNT/A1 at pH 5.5), EMD-50166 (local refinement map of SV2B-LD–BoNT/A1 on EMD-50151), EMD-50154 (local refinement map of BoNT/A1 on EMD-50151), EMD-50158 (local refinement map of SV2B-LD-H$_C$A1 on EMD-50151). EMD-50163 (local refinement map of LCH$_N$A1 on EMD-50154). The atomic coordinates have been deposited in the RCSB Protein Data Bank (PDB) under accession codes 9F25 (LCH$_N$A1 for EMD-50138), 9F1R (SV2B–H$_C$A1 for EMD-50135), 9F2B (SV2B-LD–BoNT/A1 for EMD-50146), 9F2J (SV2B-BoNT/A1 for EMD-50147 based on 9F2B and 9F1R models), 9F2Y (SV2B-LD–BoNT/A1 (pH 5.5) for EMD-50166), 9F3C (SV2B-BoNT/A1 (pH 5.5) for EMD-50151 based on 9F3C and 9F1R models). Structures used for comparative analysis in this manuscript can be found with the following PDB or EMD accession codes: 3BTA (crystal structure of BoNT/A1 holotoxin); PDB7QFP (cryo-EM structure of BoNT/E), PDB5N0B (crystal structure of TeNT), EMD-3588 (cryo-EM structure of TeNT), PDB2VU9 (crystal structure of H$_C$A1), PDB6UL6 (crystal structure of LCH$_N$A1), PDB4JRA (crystal structure of the SV2C-LDH$_C$A1 complex), PDB5JLV (crystal structure of the gSV2C-LD-H$_C$A1 complex), PDB7UIB (crystal structure of the SV2Ac-LD-H$_C$E1 complex), PDB8ET6 (cryo-EM structures of OCT1), PDB7ZH0 (cryo-EM structure of OCT3), PDB7ZH6 (cryo-EM structure of OCT3-inhibitor complex), PDB3V0B (crystal structure of the NTNHA-BoNT/A1 complex), PDB8UO8 (cryo-EM structure of SV2B) and PDB8JLF (cryo-EM structure of SV2A). Source data are provided with this paper.

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

## Acknowledgements

We thank Dr. Miroslav Peterek and Dr. Bilal Qureshi (ScopeM, ETH Zurich) for their help in cryo-EM data collection. We thank Dr. Mohamed Chami and Carola Alampi at the BioEM Lab of the Biozentrum, University of Basel, for their support with the TEM experiments. Dr. Thomas Binz (Hannover Medical School) is acknowledged for the gift of the plasmid encoding inactive BoNT/A1. We also thank Dr. Spencer Bliven, Dr. Marc Caubet Serrabou and Dr. Greta Assmann (PSI) for their support of high-performance computing. Furthermore, we acknowledge Daniel Frey (PSI) for the support with the instruments for protein purification. This work was supported by grants 310030_212253 and 198545 of the Swiss National Science Foundation to R.A.K. and V.M.K., respectively.

## Author contributions

B.K., O.L., R.A.K., and V.M.K. designed the research; B.K., O.L., R.A.K., and V.M.K. carried out the research; B.K., O.L., R.A.K., and V.M.K. analyzed the data; R.A.K. wrote the manuscript; B.K., O.L., and V.M.K. prepared the figures and co-wrote the manuscript. S.K.P. purified proteins and performed experiments for the revision.

## Competing interests

The authors declare no competing interests.
