## [Transparent Peer Review file · Nature Communications]

Cryo-EM structure of the botulinum neurotoxin A/SV2B complex and its implications for translocation

Corresponding Author: Dr Richard Kammerer

Version 0:

Reviewer comments:

Reviewer #1

(Remarks to the Author)

The major unsolved problem in the mode entry and action of Botulinum neurotoxins inside presynaptic nerve terminals is how the toxin translocates from the lumen of an intracellular acidic compartment (most likely synaptic vesicles) to the cytosol where the L domain of the toxin displays its metalloprotease activity. In the case of botulinum neurotoxin type A1 used here the carboxy-terminal region of SNAP-25 is removed by cleavage. Therefore the paper address a very important aspect of science.

The paper by Khanppnavar et al. tackles the problem of membrane translocation of BoNT/A1 using cryo-EM, a technique that in conjunction with recently developed biocomputing techniques should enable the authors to solve the structure of the intermediate steps that BoNT/A1 undergoes during this process.

This is a very appropriate technique with respect to the aim and the experimental part and analysis of data are very carried out.

The authors' aim is the more important if one considers that: i) BoNT/A1 is the second most sold pharmaceuticals and ii) it is clear that BoNTs cross membrane at low pH differently from anthrax toxins and from the other bacterial toxins that reach the cell cytosol to perform their action in the cytosol. The only one that resembles BoNT is diphtheria toxin in addition to tetanus toxin, as to say that it is likely that there are important novelties of general interest to be discovered by studying BoNT, TeNT and DT membrane crossing using cryo-EM.

In the present paper the authors, using full length protein receptor and full length BoNT/A1, not used before, as well as fragments, define the molecular anatomy of the interaction of BoNT/A1 with different isoforms of its protein receptor SV2. More in general they show that:

- a) In solution at neutral pH most molecules of BoNT/A1 adopt a semi-closed conformation resembling those of tetanus toxin. This finding corrects the current knowledge of an open conformation deduced by Lacey et al (1998) using X-ray crystallography on crystals of BoNT/A1. Among other aspects, this notion is of fundamental importance for the definition of BoNT/A1 complexes with specific neutralizing antibodies.
- b) Upon binding of BoNT/A1 to its SV2 protein receptor, in the presence of the glycolipid receptor GT1b, the toxin acquires an open conformation, which is incompatible with translocation, as it is too far from the membrane surface.
- c) At pH 5.5, receptor bound BoNTA1 changes conformation to a semi-closed conformation which the authors define as membrane translocation competent.

These very important findings qualify this paper for publication after revision. In fact, there are several points that can be corrected or improved or extended.

Major points:

- 1) Line 65 and 66: this sentence is not correct, as the interchain disulfide bond is not reduced in the cytosol, but on the cytosolic face of the membrane by the Thioredoxin-Thioredoxin Reductase system ([https://www.cell.com/action/showPdf?pii=S2211-1247\(14\)00680-9](https://www.cell.com/action/showPdf?pii=S2211-1247(14)00680-9)). This point is relevant to the understanding of the membrane translocation of the L chain from the lumen to the cytosolic face of the vesicle, as it is very likely that the bulky and hydrophobic SS bond is one of the first

parts of the toxin to be inserted in the membrane lipid bilayer.

2) In addition, correct "point 4. Translocation" in Fig. 1 A.

3) Line 83. The authors should reconsider or, better, delete their sentence "and the closely related tetanus toxin (TeNT)38 ". In fact, if one examines figures 5 and 6 of reference 38, and considers that mice were injected with 1-5 micrograms of toxin per mice (this corresponds to 50-250 microgram per Kg when the smallest value for the minimal lethal dose (MLD) reported for TeNT in mice is 0.15 nanograms/Kg (<https://pubmed.ncbi.nlm.nih.gov/31771110/>). This comparison is striking even if one considers values of MLD of 1-5 nanograms/Kg of mice that are reported for poorly active TeNT preparations). How significant can be information obtained in vivo using toxin amount several orders of magnitude higher than the MLD? An additional clearcut evidence that SV2 is not the receptor mediating the specific action CNS is reported in ref 38 that shows that TeNT does not bind inhibitory interneurons, which are the specific neuronal targets of TeNT causing tetanus.

4) Line 386. In the detailed discussion of their results with respect to previous literature, the authors fail to quote a piece of work that identifies three carboxylates residues which are involved in the membrane insertion/translocation of LC. The residues (Glu-48, Glu-653, and Asp-877 of BoNT/B) are: i) conserved among BoNTs and ii) have pKa values that indicates their protonation/neutralization at the luminal synaptic vesicle pH (<https://febs.onlinelibrary.wiley.com/doi/epdf/10.1016/j.febslet.2013.10.010>).

Minor points

1) Lines 32 and 33, please note that eight different serotypes of BoNT exists because a BoNT/X has been characterized that is not recognized by any of the seven classical antisera (Zhang, S., Masuyer, G., Zhang, J. et al. Identification and characterization of a novel botulinum neurotoxin. *Nat Commun* 8, 14130 (2017). <https://doi.org/10.1038/ncomms14130>).

2) line 38. in addition, the authors could mention that there are additional BoNT-like toxins specific for insects or BoNT-like whose specificity is not known (BoNT produced by *Weissella orizae*). This would present BoNTs as protein toxins that attack other sub-phyla others that vertebrates and give a larger view to the introduction.

3) Line 40, one more reference could be added to ref. 14 to qualify this sentence:
<https://pubmed.ncbi.nlm.nih.gov/31771110/> doi: 10.3390/toxins11120686.

4) Line 44, BoNTs and BoNT-like toxins

5) Line 52, add: and BoNT/X cut.....

Reviewer #2

(Remarks to the Author)

Here the authors utilized the cryo-EM approach and reported several interesting structures:

(1) The cryo-EM structure of full-length (detoxified) botulinum neurotoxin A (BoNT/A). The structure shows a "semi-closed" conformation (the receptor-binding domain (HcA) is close to the light chain) distinct from the "open" conformation (HcA and LC are separated and located on each side of the translocation domain) observed in previous crystal structures.

(2) Full-length SV2B in complex with HcA and another structure in complex with full-length BoNT/A. It complements a series of recent papers reporting the cryo-EM structure of full-length SV2A and SV2B for this family of important and mysterious synaptic vesicle proteins. Both the structure and mutagenesis studies confirmed that BoNT/A recognizes both SV2B protein sequence and an N-linked glycan. An interesting finding is that BoNT/A is in "open" state in the complex with SV2B (under pH8).

(3) Full-length SV2B in complex with BoNT/A under pH5.5. This new structure showed BoNT/A undergoes domain reorientation into a "semi-closed" conformation, in which the light chain and translocation domain of the toxin could be close to the membrane. This domain reorientation involves changes in the key helical linker region between HcA and the translocation domain.

This study presents highly valuable and significant insights into the structural dynamics of BoNT/A when bound to the receptor SV2 under different pH conditions. It also presents the structure of SV2B, revealing several interesting features (e.g. the disulfide bond between LL1 and LD). I recommend it for publication after a revision.

Major points:

1. The readers can benefit from a comprehensive comparison of the authors' new cryo-EM structures with recently reported full-length SV2A and SV2B, as well as SV2A-BoNT/A complexes.
2. The title, "Structural Basis of the Initial Steps of Botulinum Neurotoxin Translocation" is not proper since there is no data in this manuscript studying translocation. The focus of this manuscript is on the structure of BoNT/A, SV2B, SV2B-BoNT/A complex, and the title should be changed to reflect the actual data of the study, not authors' over-interpretation.
3. For the structural basis of the initial steps of botulinum neurotoxin translocation, as the most significant selling point of this manuscript, it lacks experimental validation of the functional implications of the observed conformational changes. To determine if the observed conformational changes are relevant to translocation, functional evidence is needed and could be for future studies.
4. Is it possible to resolve BoNT/A cryo-EM structure under low pH condition?
5. Is there a structural explanation for the conformational changes of BoNT/A upon binding to SV2 and under low pH conditions?
6. Is the disulfide bond in SV2B required for binding of BoNT/A?

Minor points:

Some figures are difficult for readers to follow. Some cryo-EM maps and structural models are too small, and the labeling is not very clear.

Abstract: the last sentence needs to be deleted.

Line 55: a lipid binding loop has also been reported by several papers to be involved for the toxin neuronal recognition.
Line 90: delete "dramatically".
Figure 1: in semi-closed state, any notable interactions between LC-Hn with Hc?
Line 200: "LDL"?
Line 205: there is no experimental evidence to say that the open state is "translocation-incompetent", authors need to tune down these claims and focus on the structure itself.
Line 209: "the adopts"?
Line 228: "which may that".
Line 231-234: any experimental dataset? Otherwise move it to Discussion.
Line 246: "HCA".
Check references, many missing page numbers.
Line 311: "comparison with NTNH" should be in complex with NTNH.
Line 408-410: what do you mean by comparable to the full-length toxin?
Line 494: "HCA1"
What are the pH conditions for BoNT/A and for SV2B-HcA?
The method section mentioned SV2B-BoNT/A1-GT1b complex, which was not described in the main text?
The supplementary information section overall is not precisely written. For example:
Line 185: "The cryo-EM dataset of BoNT/A1 composed of 8'014 movies was acquired with a total dose of 80 e/Å²." Such a high total dose during data collection likely results in the loss of high-resolution structural information. Such 80 e/Å² high total dose used here needs clarification.
Line 189: "The remaining 8'014 micrographs were subjected to automated particle picking using the Laplacian-of-Gaussian program in RELION 3.1.3 to obtain the initial particle set." It is very uncommon for all 8,014 movies to have an estimated CTF resolution better than 4 Å.
Line 221: "The detailed steps of cryo-EM processing are depicted in Figure S*." The specific figure numbers are consistently not mentioned in the draft.

Version 1:

Reviewer comments:

Reviewer #1

(Remarks to the Author)

I have read the revised version and the authors' letter of response to the points raised by the Reviewers. It appears to me that they have responded adequately to almost all the points raised by the Reviewers and that the revised paper has been largely improved. I also that they have done their best to answer to the most relevant point, i.e. the one raised by Reviewer 2 on the attempt to acquire the structure of bound BoNT after lowering the pH.

Reviewer #2

(Remarks to the Author)

I appreciate authors' efforts in revising the manuscript and I am satisfied with their work.

Minor suggestions:

1. Title: a title clearly indicating that the paper is on the structure analysis of BoNT/A-SV2B complexes, something like: "structure of botulinum neurotoxin A in complex with full-length SV2B shows ..." might be more straightforward and can stand the test of time, but I leave this to the discussion between the authors and editors.
2. Line 187: "SV2B-HCA1", is this correct?

Reviewer #3

(Remarks to the Author)

In this manuscript, the authors use cryo-EM to solve structures of botulinum toxin (BoNT/A1) alone and in complex with its receptor synaptic glycoprotein 2B (SV2B). I was asked to evaluate the technical quality of the cryo-EM work and model building.

Overall, I think the authors have done a nice job outlining how they have processed the data and improved the quality of their maps using focused refinements. However, I think they should address the following concerns:

- 1) I think the figures are all too small. I can't really see any of the structural details that are being highlighted, especially for Figures 2-5. I think Figure 6 is nice.
- 2) Please deposit the associated masks that were used to generate all focused refinement maps.

3) Generally, I think all the models in this manuscript are overbuilt. While I agree with the position of the backbone for most of their structures, they have modeled side chains in areas where there is simply no density to support the chosen rotamer. How did the authors decide to position side chains? I think the authors have avoided making conclusions that hinge on the position of side chains, but when readers download maps for molecular docking studies or molecular dynamics simulations, they often do not realize that the rotameric positions for side chains are not certain. I personally would stub the side chains when there is no density for positioning rotamers.

In addition, I noticed that the authors (seemingly randomly) added two rotameric positions for residue Asp 131 in the BoNT/A1 model. There is no evidence to support this. I was unable to look at every side chain for every model in the manuscript. If there are other examples of this in their models, it should be corrected/removed. Of note, the only major conclusion the authors make related to specific amino acids (and sugar groups) is represented in Figure 3, at the SV2B-BoNT/A1 binding interface and the density here looks quite good. The position of the residues is supported by the density in the maps and their conclusions are solid for this part of the manuscript.

4) The authors describe that they made inactivating mutants E224Q, R362A and Y365F in the methods section. However, in their models, residue E224Q is present but the other two mutations are represented in the models as R363A and Y366F. Where is the error? Is the methods section written incorrectly or is the register of their models wrong? This needs to be fixed.

5) For the SV2B-HcA1 data, the authors have left out residues in their model from residue 330 to residue 369. However, there is some unmodeled density near residue 329 in chain A and near residue 370 in chain A. The density is quite strong, especially near residue 370, and is at least as strong as the density elsewhere in their map. Did the authors attempt to model this density? What could this density represent? It seems too strong to simply be noise.

6) For the SV2B-HcA1 data, the authors have left out residues in their model from residue 1211 to residue 1216 in chain B. However, there is some unmodeled density in the gap. Did the authors attempt to model this density to fill in the model?

7) For the pH 5.5 (or pH 5.0?) data, the overall map is not of high quality. The authors do improve the maps when they performed focused refinements, which is good. I want to stress again that the authors should provide their masks along with the maps. Despite the improvement from focused refinements, I think the models for this data are overbuilt. There simply is not density to support most of the side chains. I think the authors need to put in a clear and obvious disclaimer that there is confidence only in the positions of the backbone atoms (especially for the pH 5.5 data) and not the side chains. Alternatively, they should simply stub the sidechains.

POINT-BY-POINT RESPONSE TO REVIEWER'S COMMENTS

We thank both reviewers for their time to review the manuscript. Their comments and suggestions helped to further improve the quality of the paper.

Reviewer #1:

Major points:

1) Line 65 and 66: this sentence is not correct, as the interchain disulfide bond is not reduced in the cytosol, but on the cytosolic face of the membrane by the Thioredoxin-Thioredoxin Reductase system ([https://www.cell.com/action/showPdf?pii=S2211-1247\(14\)00680-9](https://www.cell.com/action/showPdf?pii=S2211-1247(14)00680-9)). This point is relevant to the understanding of the membrane translocation of the L chain from the lumen to the cytosolic face of the vesicle, as it is very likely that the bulky and hydrophobic SS bond is one of the first parts of the toxin to be inserted in the membrane lipid bilayer.

Response. In the revised version of the manuscript, we replaced the sentence "Once reaching the reducing environment of the cytosol, the disulfide bond connecting LC and HC is reduced, resulting in the release of the protease." by "Once reaching the cytosolic surface on the synaptic vesicle membrane, the disulfide bond connecting LC and HC is reduced by the thioredoxin-thioredoxin reductase system, resulting in the release of the protease." Furthermore, we included the Cell Reports reference mentioned by the reviewer.

2) In addition, correct "point 4. Translocation" in Fig. 1 A.

Response: As suggested by the reviewer, we modified "point 4. Translocation" of Fig. 1A. We indicated the disulfide bridge in Fig. 1a and added the following sentence to the figure legend: "LC translocation and reduction of the disulfide bridge connecting the LC and HC on the cytosolic face of the synaptic vesicle membrane by the thioredoxin-thioredoxin reductase system"

3) Line 83. The authors should reconsider or, better, delete their sentence "and the closely related tetanus toxin (TeNT)³⁸". In fact, if one examines figures 5 and 6 of reference 38, and considers that mice were injected with 1-5 micrograms of toxin per mice (this corresponds to 50-250 microgram per Kg when the smallest value for the minimal lethal dose (MLD) reported for TeNT in mice is 0.15 nanograms/Kg (<https://pubmed.ncbi.nlm.nih.gov/31771110/>)). This comparison is striking even if one considers values of MLD of 1-5 nanograms/Kg of mice that are reported for poorly active TeNT preparations). How significant can be information obtained in vivo using toxin amount several orders of magnitude higher than the MLD? An additional clearcut evidence that SV2 is not the receptor mediating the specific action CNS is reported in ref 38 that shows that TeNT does not bind inhibitory interneurons, which are the specific neuronal targets of TeNT causing tetanus.

Response: We deleted the last part of the sentence "and the closely related tetanus toxin (TeNT)³⁸".

4) Line 386. In the detailed discussion of their results with respect to previous literature, the authors fail to quote a piece of work that identifies three carboxylates residues which are involved in the membrane insertion/translocation of LC. The residues (Glu-48, Glu-653, and Asp-877 of BoNT/B) are: i) conserved among BoNTs and ii) have pKa values that indicates their protonation/neutralization at the luminal synaptic vesicle pH (<https://febs.onlinelibrary.wiley.com/doi/epdf/10.1016/j.febslet.2013.10.010>).

Response: We thank the reviewer for bringing this study to our attention. It nicely supports our findings. We added the following two sentences to the revised version of the manuscript: "Our

conclusions are consistent with the results of a mutational study of three conserved negatively-charged amino acids in BoNT/B (E48 and E653 to Q; D877 to N) whose side chains are predicted to be protonated at physiological synaptic vesicular pH. The triple mutant showed increased neurotoxicity as a result of faster LC translocation, indicating that neutralisation of specific negative surface charges facilitates the interaction of the toxin with the synaptic vesicle membrane.” The reference reporting the results was added to the manuscript.

Minor points:

1) Lines 32 and 33, please note that eight different serotypes of BoNT exists because a BoNT/X has been characterized that is not recognized by any of the seven classical antisera (Zhang, S., Masuyer, G., Zhang, J. et al. Identification and characterization of a novel botulinum neurotoxin. Nat Commun 8, 14130 (2017). <https://doi.org/10.1038/ncomms14130>).

Response: We include BoNT/X in the revised introduction of the manuscript. We rephrased the sentence to: “They are divided into seven classical serotypes, designated BoNT/A - BoNT/G, and the recently identified BoNT/X that are all mainly produced by the gram-positive bacterium *Clostridium botulinum*.” The reference reporting the characterization of BoNT/X was added to the manuscript.

2) line 38. in addition, the authors could mention that there are additional BoNT-like toxins specific for insects or BoNT-like whose specificity is not known (BoNT produced by *Weissella oryzae*). This would present BoNTs as protein toxins that attack other sub-phyla others that vertebrates and give a larger view to the introduction.

Response: We thank the reviewer for this suggestion, which provides important additional information to the introduction. We added the following two sentences to the manuscript: “Furthermore, three non-clostridial BoNT-like proteins, termed BoNT/Wo, BoNT/En and PMP1, have recently been identified in *Weissella oryzae* SG25T, *Enterococcus faecium* and *Paraclostridium bif fermentans*, respectively. Notably, PMP1 is specific for insects while the targets of BoNT/Wo and BoNT/En are unknown.” The references reporting the characterization of the toxins were added to the manuscript.

3) Line 40, one more reference could be added to ref. 14 to qualify this sentence: <https://pubmed.ncbi.nlm.nih.gov/31771110/>) doi: 10.3390/toxins11120686.

Response: We added the reference suggested by the reviewer to the manuscript.

4) Line 44, BoNTs and BoNT-like toxins

Response: Done

5) Line 52, add: and BoNT/X cut.....

Response: We changed “Serotypes BoNT/A and /E cleave synaptosomal-associated protein 25 (SNAP-25), whereas BoNT/B, /D, /F and /G cut vesicle-associated membrane protein (VAMP). Only BoNT/C is known to cleave two substrates, SNAP-25 and syntaxin.” to “Serotypes BoNT/A, and E cleave synaptosomal-associated protein 25 (SNAP-25), BoNTs B, D, F, G, Wo and X cut vesicle-associated membrane protein (VAMP) and PMP1 cleaves syntaxin. BoNT/C and BoNT/En can cleave two substrates, SNAP-25 and syntaxin and SNAP-25 and VAMP, respectively.” The additional references reporting these findings were added to the manuscript. Furthermore, we also included the following sentence in the Introduction section: “The cell-surface receptors of BoNT/X, BoNT/En, PMP1 and

BoNT/Wo have not yet been identified, although the first three proteins contain a predicted PSG-binding site.”

Reviewer #2:

Major points:

1. The readers can benefit from a comprehensive comparison of the authors’ new cryo-EM structures with recently reported full-length SV2A and SV2B, as well as SV2A-BoNT/A complexes.

Response: As suggested by the reviewer, we compared our SV2B cryo-EM structure to the recently solved cryo-EM structures of SV2A (Yamagata et al., Nature Communications, 2024) and SV2B (Mittal et al., Nature Structural and Molecular Biology, 2024). We included the comparison in an additional Supplementary Figure (Supplementary Fig. 7). Furthermore, we added the following text to the section “Cryo-EM structure of the SV2B-BoNT/A1 complex” of the Results: “While we were submitting this manuscript, the cryo-EM structures of SV2A and SV2B were reported^{55,56}. Overall, they are very similar to our structure. A comparison of our SV2B-H_cA1 cryo-EM structure to the published receptor structures is shown in the Supplementary Fig. 7.”

Supplementary Fig. 7. Comparison of our SV2B cryo-EM structure with the published SV2A and SV2B cryo-EM structures. A, B Overlay of our SV2B structure (orange) with recently determined structures of SV2B (blue, PDB ID: 8UO8) and SV2A (magenta, PDB ID: 8JLF). The root mean square deviation (RMSD) based on an all-atom alignment of our SV2B structure and the published SV2B and SV2A structures is 0.93 Å and 1.81 Å, respectively.

2. The title, "Structural Basis of the Initial Steps of Botulinum Neurotoxin Translocation" is not proper since there is no data in this manuscript studying translocation. The focus of this manuscript is on the structure of BoNT/A, SV2B, SV2B-BoNT/A complex, and the title should be changed to reflect the actual data of the study, not authors' over-interpretation.

Response: We changed the title to "Receptor-bound botulinum neurotoxin A adopts a translocation-competent conformation at low pH".

3. For the structural basis of the initial steps of botulinum neurotoxin translocation, as the most significant selling point of this manuscript, it lacks experimental validation of the functional implications of the observed conformational changes. To determine if the observed conformational changes are relevant to translocation, functional evidence is needed and could be for future studies.

Response: We are grateful to the reviewer for pointing this out. We believe the manuscript in its current form provides a very clear structural framework for developing an in-depth understanding of the early steps in toxin recognition by its receptor, and the subsequent pH-dependent conformational transitions concomitant with synaptic vesicle acidification. These changes presumably prime the toxin for subsequent translocation. The reviewer is absolutely correct in that we will need eventually to validate these observations experimentally in relation to the different stages of toxin translocation. However, the key problem in the field at the moment is that the molecular mechanisms of toxin translocation are not yet clearly defined: there is no clarity on the exact mode of toxin translocation across the membrane. With this in mind, we believe that a full validation of our observation will require first establishing a solid experimental framework for studying the actual steps of toxin translocation – this is our goal for near and long-term future, and we intend to pursue this vigorously.

4. Is it possible to resolve BoNT/A cryo-EM structure under low pH condition?

Response: Following the reviewer's suggestion, we attempted to determine the cryo-EM structure of BoNT/A1 at low pH. We prepared the protein and the cryo-EM grids, collected a substantial cryo-EM dataset of 14'618 movies and performed image processing using the established procedure (similar to that used for the toxin at higher pH). From 11'657 micrographs with a CTF better than 5Å, around ~4 million particles were autopicked, and after multiple rounds of 2D classifications, only 135'486 particles showed reasonable features for BoNT/A1 (Response Fig. R1A). The 2D classes and a view of the best 3D class are shown in the Response Fig. R1. The 2D classes showed secondary structure elements but were not as well defined as the 2D classes of the toxin at higher pH, indicative of destabilization of the particles. Moreover, ab-initio 3D reconstructions revealed heterogeneity of BoNT/A1, with distinct populations of the protein (Fig. R1B). One of the 3D classes (44.3 % of the total particles used for 3D reconstruction after 2D classification) showed features consistent with a semi-closed state observed in receptor-unbound and receptor-bound low pH conditions (Fig. R1C). However, based on this low-resolution information we are not able to draw firm conclusions yet.

A detailed understanding the pH-dependent conformational transitions of BoNT/A1 will require a separate in-depth investigation, integrating not only cryo-EM data, but EPR, single molecule fluorescence, simulations, a variety of other biophysical techniques and comparisons of more than one toxin, all this will fall far outside the scope of this study. Therefore, we would strongly prefer to

not include this data into the manuscript, to have an opportunity to do a more comprehensive and robust analysis in the future.

Fig. R1. Cryo-EM analysis of BoNT/A1 at pH 5.5. **A** 2D classification of BoNT/A1 particles extracted from a dataset of 14'618 micrographs (total). **B** 3D classes obtained by ab-initio reconstruction in cryo-SPARC indicate a single class (~44% of the particles after 2D classification), which is consistent with the toxin conformation. **C** The Class 1 density map, overlaid with our BoNT/A1 model (magenta) in two different orientations. The positions of the key elements of toxin structure are identifiable in the density map, although the quality of the map is substantially lower compared to the one of BoNT/A1 at pH 8. The C-terminal subdomain of H_CA1 is only very poorly defined (marked with an asterisk).

5. Is there a structural explanation for the conformational changes of BoNT/A upon binding to SV2 and under low pH conditions?

Response: The helical linker at the H_C-H_N interface clearly plays a key role in the conformational changes of the toxin. But the helical linker alone is not sufficient because at neutral pH the unbound and the receptor bound toxin adopt a semi-closed and an open conformation, respectively. Receptor and toxin have an overall negative charge at neutral pH. Therefore, binding of the negatively-charged

toxin to the even more negatively-charged receptor might cause the conformational change through electrostatic repulsion.

6. Is the disulfide bond in SV2B required for binding of BoNT/A?

Response: We performed MST experiments under reducing conditions using SV2B and H_CA1. The quality of the measurements is not as good as under non-reducing conditions. Reduction of the disulfide bridge in SV2B results to some extent in precipitation, which is probably the result of protein destabilization. However, the toxin domain still binds to the receptor with a similar K_D value as under non-reducing conditions. This finding is also supported by the availability of several complex crystal structures of the receptor binding domain of BoNT/A (H_CA) to the isolated luminal domain of SV2C (SV2C-LD). Because reduction of the disulfide bridge of SV2B is very likely to represent a non-physiological situation, we would prefer not to add these results to the manuscript.

Fig. R2. Microscale thermophoresis-based binding assay.

Minor points:

Some figures are difficult for readers to follow. Some cryo-EM maps and structural models are too small, and the labeling is not very clear.

Response: We updated all main figures, increasing the sizes of the labels and the elements that previously came across as too small / crowded. The changes included splitting Fig. 3 into two figures to improve the quality of figure presentation. We additionally increased the sizes of the 2D classes shown in the description of the cryo-EM part, which previously likely lead to the observed effect of various elements being too small. We hope these changes are acceptable.

Abstract: the last sentence needs to be deleted.

Response: As suggested by the reviewer, we deleted the last sentence of the Abstract.

Line 55: a lipid binding loop has also been reported by several papers to be involved for the toxin neuronal recognition.

Response: We added this finding and also mention that N-glycosylation and probably also a lipid-binding site in the N-terminal subdomain of H_CA contribute to toxin-receptor binding. "Furthermore, a lipid-binding loop was recently identified in BoNTs B, C, D, DC and G that appears essential for the

potency of the toxins. A lipid-binding site was also identified in the N-terminal half of H_CA. Moreover, it was shown that N-glycosylation of SV2 contributes to BoNT/A binding. A multiple interaction model with receptors or particular posttranslational modifications that have moderate toxin affinity would provide a plausible explanation why BoNTs are so extremely toxic at very low concentrations and why they are highly specific for predominantly cholinergic nerve terminals.” The references reporting the findings were added to the manuscript.

Line 90: delete “dramatically”.

Response: We deleted “dramatically”.

Figure 1: in semi-closed state, any notable interactions between LC-H_N with H_C?

Response: The density map of BoNT/A1 in its semi-closed conformation is limited (as shown Fig. R3A, B and also depicted in Supplementary Figs. 2 and 3), making it challenging to accurately define specific side-chain interactions. Therefore, we would rather avoid strong statements about specific interactions between the LC-H_N and H_C. From the best model of BoNT/A1 in a semi-closed state that was obtained in the presence of SV2 and at pH 5.5, it is possible that residues R393-N394 and N1245-D1246 are in close proximity (3.5-4.3Å), however the exact distances between the side chain atoms cannot be determined at this resolution as evident from the features of the density maps (Fig. R3B).

Fig. R3A, B Overlay of BoNT/A1 on the cryo-EM density map (contoured at sigma threshold levels 5σ and 8σ) showing poorly defined density features for H_C . **C, D** The receptor-bound BoNT/A1 in its semi-closed conformation at low pH. Residues in H_C and LC within 6 \AA of the neighboring domain are shown as sticks. The $C\alpha$ atom distances for residue pairs R393-D1246 and N394-N1245 are indicated with the dotted lines. These close distances position the side chains of these residues within $3.5\text{-}4.3\text{ \AA}$ of each other, however the exact distances between the side chain atoms cannot be determined at this resolution (evident from the features of the density maps (contoured at sigma threshold levels 8σ and 5σ) shown in panel D).

Line 200: “LDL”?

Response: The mistake was corrected. The nomenclature of the linkers connecting LD with the TM part was rather confusing. In the revised manuscript, we therefore omitted specific names for the linkers and refer to the β -helix domain and the linkers as LD.

Line 205: there is no experimental evidence to say that the open state is “translocation-incompetent”, authors need to tune down these claims and focus on the structure itself.

Response: We removed “translocation-incompetent”. Overall, we also tuned down these statements.

Line 209: “the adopts”?

Response: We corrected the mistake and added the missing word “toxin” (“the toxin adopts”).

Line 228: “which may that”.

Response: We apologize for the annoying mistake. This part of the sentence should read “which may be the reason that”.

Line 231-234: any experimental dataset? Otherwise move it to Discussion.

Response: We removed lines 231-234 from the manuscript.

Line 246: “HCA”.

Response: We corrected “HCA” to “H_CA”

Check references, many missing page numbers.

Response: The reference list was generated with EndNote. Many journals that publish exclusively online like Scientific Reports or Nature Communications don't provide specific page numbers.

Line 311: “comparison with NTN_H” should be in complex with NTN_H.

Response: We corrected this part of the sentence to “Comparison to BoNT/A1 in complex with NTN_HA”.

Line 408-410: what do you mean by comparable to the full-length toxin?

Response: These results were obtained by Fischer and colleagues (Fischer A. et al., Botulinum neurotoxin devoid of receptor binding domain translocates active protease. PLoS Pathog. 4, e1000245 (2008)). They write: “The extent of proteolysis attained by LC-TD was comparable to that produced by holotoxin, albeit it required higher protein concentration consistent with a lower efficacy.” In their paper, they show that full-length BoNT/A1 cleaved approximately 60% of SNAP-25 of Neuro 2A cells when incubated at a concentration of 5 μ g/well (Figure 5). In contrast, BoNT/A1 devoid of the H_C (LC-TD) cleaved more >40% and >80% of SNAP-25 at concentrations of 5 and 30 μ g/well, respectively. For clarity, we now write “at an efficiency that was lower than the full-length toxin.”

Line 494: “HCA1”

Response: Corrected.

What are the pH conditions for BoNT/A and for SV2B-HcA?

Response: The buffer / pH conditions were as defined in the Methods section under "Cryo-EM sample preparation": BoNT/A1, pH 7.4; SV2B-HcA, pH 8.0. "Final protein concentrations were 0.4-0.5 mg/ml for BoNT/A1 (pH 7.4), 4-5 mg/ml for SV2B-HcA1 (pH 8.0), 2.5-3.5 mg/ml for SV2B-BoNT/A1-GT1b (pH 8.0) and 3-4 mg/ml for SV2B-BoNT/A1 (pH 5.5)."

The method section mentioned SV2B-BoNT/A1-GT1b complex, which was not described in the main text?

Response: We have clarified this point in the section "Cryo-EM structure of the SV2B-BoNT/A1 complex" of the Results: "To elucidate how SV2B interacts with the toxin, we prepared SV2B complexes with the inactive full-length BoNT/A1 (in the presence of a 20 molar excess of ganglioside GT1b) and HcA1." and "Although applied in excess, GT1b was not detectable in the complex structure with the full-length toxin."

The supplementary information section overall is not precisely written. For example:

Line 185: "The cryo-EM dataset of BoNT/A1 composed of 8'014 movies was acquired with a total dose of 80 e/Å²." Such a high total dose during data collection likely results in the loss of high-resolution structural information. Such 80 e/Å² high total dose used here needs clarification.

Response: We tried to improve the entire Supplementary Information section and specifically addressed the points raised by the reviewer. We hope that the new version is acceptable.

We used a relatively high dose of 80 e/Å² to enhance the contrast of the relatively small-sized and conformationally dynamic BoNT/A1 particles. The dose fractionation implemented approach in RELION-3.1.3. was used, maximizing the contrast by using low resolution information across all movie frames, while minimizing the loss of high-resolution information (which is preserved in the early frames, before the high dose is accumulated). We have now clarified this by adding the following sentence (page 5 in the supplementary information): "The cryo-EM dataset of BoNT/A1 composed of 8'361 movies was acquired with a total dose of 80 e/Å². The dose was chosen to enhance the contrast of the small and conformationally dynamic BoNT/A1 particles."

Line 189: "The remaining 8'014 micrographs were subjected to automated particle picking using the Laplacian-of-Gaussian program in RELION 3.1.3 to obtain the initial particle set." It is very uncommon for all 8,014 movies to have an estimated CTF resolution better than 4 Å.

Response: We thank the reviewer for noticing this mistake. The total number of movies collected for the BoNT/A1 sample was 8'361. 8'014 micrographs were estimated to have a better resolution than 4Å by Gctf. This has been updated in the revised manuscript.

Line 221: "The detailed steps of cryo-EM processing are depicted in Figure S*." The specific figure numbers are consistently not mentioned in the draft.

Response: We apologize for this careless mistake. We have updated the manuscript with the correct figure references.

POINT-BY-POINT RESPONSE TO REVIEWERS' COMMENTS

We thank the reviewers for all their suggestions and comments, which helped to improve the manuscript.

Reviewer #2

Minor suggestions:

1. Title: a title clearly indicating that the paper is on the structure analysis of BoNT/A-SV2B complexes, something like: “structure of botulinum neurotoxin A in complex with full-length SV2B shows ...” might be more straightforward and can stand the test of time, but I leave this to the discussion between the authors and editors.

Response: Since the X-ray crystal structure of BoNT/E was reported, which revealed a different conformation compared to the one of BoNT/A (Kumaran et al., domain organization in Clostridium botulinum neurotoxin type E is unique: its implication in faster translocation. J. Mol. Biol., 2009, 233-45. doi: 10.1016/j.jmb.2008.12.027) toxin conformation was always linked to translocation. In the Kumaran et al. paper translocation was even mentioned in the title, despite there existed no evidence at that time for a correlation between toxin conformation and translocation speed. Therefore, we feel that the title describes our findings in an adequate manner. However, an alternative title that as suggested by the reviewer covers structure analysis could be: “Cryo-EM structure of the botulinum neurotoxin A/SV2B complex and its implications for translocation”.

2. Line 187: “SV2B-HCA1”, is this correct?

Response: We corrected the mistake in the revised version of the manuscript.

Reviewer #3

1) I think the figures are all too small. I can't really see any of the structural details that are being highlighted, especially for Figures 2-5. I think Figure 6 is nice.

Response: We increased the sizes of the figure panels by splitting the figures into additional figures, and we now have in total 9 main figures.

2) Please deposit the associated masks that were used to generate all focused refinement maps.

Response: We already deposited all masks and half maps associated for each PDB/EMDB submission but we did not include them in the files for the reviewer. For completeness, please find all the files in the shared dropbox folder (available on request for the reviewer).

3) Generally, I think all the models in this manuscript are overbuilt. While I agree with the position of the backbone for most of their structures, they have modeled side chains in areas where there is simply no density to support the chosen rotamer. How did the authors decide to position side chains? I think the authors have avoided making conclusions that hinge on the position of side chains, but when readers download maps for molecular docking studies or molecular dynamics simulations, they often do not realize that the rotameric positions for side chains are not certain. I personally would stub the side chains when there is no density for positioning rotamers.

Response: We agree with reviewer that it is imperative that the reader (or the casual structure enthusiast who downloads the models) is not misled. As suggested by the reviewer under point 7, we want to achieve this by adding a disclaimer to the Methods section stating that the models are based on EM densities of limited resolution. Furthermore, we have modified the title of our PDB entry to “Low-resolution (5.4 Angstroms) cryo-EM structure of SV2B-BoNT/A1 at pH 5.5”. This implies that the model is of limited resolution. By definition, the 5.4 Å resolution map implies low confidence in the derived model. Since the model building today is routinely facilitated by AI-based tools (such as AlphaFold), we feel that further disclaimers in the PDB/EMDB entries are not necessary.

In those cases where the maps showed no clear side-chain densities, the specific rotamers of the side chains were assigned based on the available high-resolution crystal structure of BoNT/A1 (PDB6UL6) or on the AlphaFold model of SV2B. Introducing stubbed side chains for the whole model of the toxin would not quite reflect the quality of the better parts of the map, where we see clear side-chain densities. If we generate a mixed model with full side chains and stubbed side chains, we risk causing more confusion. Our preferred method of handling this is as mentioned above to add the following disclaimer about the quality of the maps / models in the Methods section, thereby making the viewers aware of the pitfalls of over-interpretation.

“Model building, refinement and validation

The starting coordinates for model building of the SV2B-H_cA1 complex was derived from an AlphaFold model ⁷⁸ of SV2B (AF-Q7L1I2-F1) and the previously solved X-ray crystal structure of H_cA1 (PDB2VU9). The coordinates for LCH_NA1 or BoNT/A1 modelling were obtained from a previously solved X-ray crystal structure (PDB6UL6). These initial models were then docked into the cryo-EM maps and subjected to flexible fitting with the “all-atom refine” sub-module in COOT ⁷⁹, followed by several iterative rounds of manual readjustments of the side chains in COOT. The carbohydrates in SV2B were added using the Glyco module in COOT ^{79,80}. Finally, refinement of the rebuilt model against the EM map was carried out with phenix.real_space_refine ⁸¹. These refined models of SV2B-H_cA1 and BoNT/A1 were subsequently used for generating the SV2B-BoNT/A1 complex models. All the models were validated by the Molprobtity program in PHENIX ⁸². Data collection, processing and model refinement statistics are provided in Supplementary Table 1. Because the models of BoNT/A1, SV2B-BoNT/A1 (pH 8) and SV2B-BoNT/A1 (pH 5.5) are based on cryo-EM maps of limited resolution, the precise positions of the side-chain atoms in many instances are not well defined.”

In addition, I noticed that the authors (seemingly randomly) added two rotameric positions for residue Asp 131 in the BoNT/A1 model. There is no evidence to support this. I was unable to look at every side chain for every model in the manuscript. If there are other examples of this in their models, it should be corrected/removed. Of note, the only major conclusions the authors make related to specific amino acids (and sugar groups) is represented in Figure 3, at the SV2B-BoNT/A1 binding interface and the density here looks quite good. The position of the residues is supported by the density in the maps and their conclusions are solid for this part of the manuscript.

Response: We deleted the alternative rotamer in the BoNT/A1 model. The two rotamers of D131 are present in the X-ray crystal structure (PDB6UL6), which was used for model building. It was overlooked by us during deposition.

4) The authors describe that they made inactivating mutants E224Q, R362A and Y365F in the methods section. However, in their models, residue E224Q is present but the other two mutations are represented in the models as R363A and Y366F. Where is the error? Is the methods section written incorrectly or is the register of their models wrong? This needs to be fixed.

Response: The inactivating mutations are E224Q, R363A, and Y366F. We have corrected this mistake in the Methods section of the revised manuscript.

5) For the SV2B-HcA1 data, the authors have left out residues in their model from residue 330 to residue 369. However, there is some unmodeled density near residue 329 in chain A and near residue 370 in chain A. The density is quite strong, especially near residue 370, and is at least as strong as the density elsewhere in their map. Did the authors attempt to model this density? What could this density represent? It seems too strong to simply be noise.

Response: Initially, we attempted to model this region, including residues 330-333 and 370-374. However, we observed (i) a lack of clear side chain densities in this region, (ii) a higher clash score of the resulting models, (iii) and an apparent instability of the model corresponding to this region during real space refinement in Phenix. The quality of this region of the map prompted us to omit this region from the final model.

6) For the SV2B-HcA1 data, the authors have left out residues in their model from residue 1211 to residue 1216 in chain B. However, there is some unmodeled density in the gap. Did the authors attempt to model this density to fill in the model?

Response: We avoided building the model into this region due lower confidence in the EM map. Although it is possible nowadays to generate a passable AlphaFold model that will roughly match a poor region of the map, we felt that the density map is just not robust enough for building a model of this region based on experimental data.

7) For the pH 5.5 (or pH 5.0?) data, the overall map is not of high quality. The authors do improve the maps when they performed focused refinements, which is good. I want to stress again that the authors should provide their masks along with the maps. Despite the improvement from focused refinements, I think the models for this data are overbuilt. There simply is not density to support most of the side chains. I think the authors need to put in a clear and obvious disclaimer that there is confidence only in the positions of the backbone atoms (especially for the pH 5.5 data) and not the side chains. Alternatively, they should simply stub the sidechains.

As already mentioned in the response to point 3, we have modified the title of our PDB entry to “Low-resolution (5.4 Å) cryo-EM structure of SV2B-BoNT/A1 at pH 5.5“. Furthermore, we additionally included a statement in the Methods section, in the paragraph describing model building.